# Assessing the response of groundwater quantity and travel time distribution to 1.5, 2 and 3 degrees global warming in a mesoscale central German basin

Miao Jing[1,2,3], Rohini Kumar[1], Falk Heße[1], Stephan Thober[1], Oldrich Rakovec[1,4], Luis Samaniego[1], and Sabine Attinger[1,5]

[1]Department of Computational Hydrosystems, UFZ – Helmholtz Centre for Environmental Research, Permoserstr. 15, 04318 Leipzig, Germany
[2]Institute of Geosciences, Friedrich Schiller University Jena, Burgweg 11, 07749 Jena, Germany
[3]State Key Laboratory of Geomechanics and Geotechnical Engineering, Institute of Rock and Soil Mechanics, Chinese Academy of Sciences, 430071 Wuhan, China
[4]Czech University of Life Sciences, Faculty of Environmental Sciences, Prague, 169 00, Czech Republic
[5]Institute of Earth and Environmental Sciences, University of Potsdam, Karl-Liebknecht-Str. 24–25, 14476 Potsdam, Germany

**Correspondence:** Miao Jing (mjing@whrsm.ac.cn); Falk Heße (falk.hesse@ufz.de); Rohini Kumar (rohini.kumar@ufz.de)

**Abstract.** Groundwater is the biggest single source of high-quality freshwater worldwide, which is also continuously threatened by the changing climate. In this paper, we investigate the response of the regional groundwater system to climate change under three global warming levels (1.5, 2, and 3 °C) in a central German basin (Nägelstedt). This investigation is conducted by deploying an integrated modeling workflow that consists of a mesoscale Hydrologic Model (mHM) and a fully-distributed groundwater model OpenGeoSys (OGS). mHM is forced with climate simulations of five general circulation models under three representative concentration pathways. The diffuse recharges estimated by mHM are used as boundary forcings to the OGS groundwater model to compute changes in groundwater levels and travel time distributions. Simulation results indicate that groundwater recharges and levels are expected to increase slightly under future climate scenarios. Meanwhile, the mean travel time is expected to decrease compared to the historical average. However, the ensemble simulations do not all agree on the sign of relative change. Changes in mean travel time exhibit a larger variability than those in groundwater levels. The ensemble simulations do not show a systematic relationship between the projected change (in both groundwater levels and travel times) and the warming level, but they indicate an increased variability in projected changes with the enhanced warming level from 1.5 to 3 °C. Correspondingly, it is highly recommended to restrain the trend of global warming.

## 1 Introduction

The availability, sustainability, and quality of water resources are threatened by many sources, among which the changing climate plays a critical part (Stocker, 2014). A significant sign of climate change is global warming, which has been evidenced by the analysis of long-term air temperature records (Masson-Delmotte et al., 2018). Not only the earth's surface temperature shows a constant warming trend, but the sea surface temperature has also increased (Stocker, 2014). There has been adequate

proof that the massive greenhouse gas emissions since the eighteenth century accelerate the global warming process (Stocker, 2014). Consequently, it is urgently needed to estimate the change of meteorological variables (e.g., precipitation and temperature) in the future global warming scenarios. General circulation models (GCMs) combined with different emission scenarios or representative concentration pathways (RCPs) have been widely employed for climate impact study (Masson-Delmotte et al., 2018; Collins et al., 2013; Thober et al., 2018; Marx et al., 2018).

Climate change may significantly alter the pattern of terrestrial hydrological processes, influence the spatial and temporal behavior of shallow water storages, and manipulate the degree and frequency of extreme events such as floods and droughts (Van Roosmalen et al., 2009; Sridhar et al., 2017; Thober et al., 2018; Marx et al., 2018). Hydrological processes and states (e.g., evapotranspiration, soil moisture, and potential recharge) are tightly coupled with the climate variables (e.g., precipitation, humidity, atmosphere temperature). The impact of climate change on the terrestrial water cycle is uncertain. Climate model projections show a good consistency in future global averaged trends but may disagree on the magnitude of regional-scale variables, particularly for precipitation projection (Meehl et al., 2007). Many studies estimate the control and uncertainty of climate change on hydrological states and fluxes (Hunt et al., 2013; Samaniego et al., 2018; Renée Brooks et al., 2010; Hattermann et al., 2017; Goderniaux et al., 2009). The frequency and intensity of extreme events (e.g., soil moisture drought, heatwave) may be exacerbated owing to anthropogenic warming (Samaniego et al., 2018; Kang and Eltahir, 2018; Marx et al., 2018). The global water scarcity is likely to be exacerbated due to the potential decline in freshwater resources under the 2 °C global warming level (Schewe et al., 2014; Singh and Kumar, 2019; Gosling et al., 2017).

As the single biggest source of the world's fresh water supply, groundwater plays a critical role in the sustainability of the terrestrial ecosystem and the environmental consequences of climate variability. Globally, groundwater makes up 35% of the total freshwater withdrawals, constituting approximately 36%, 27% and 42% of water consumption for households, manufacturing, and agriculture, respectively (Döll et al., 2012). Although the general knowledge of scale-dependent hydraulic properties of the subsurface hydrologic systems is still quite limited, they prove to be increasingly influenced by anthropogenic factors (Küsel et al., 2016). The worldwide groundwater system can be affected by climate variability directly by a change in recharge or indirectly by a change in groundwater abstraction (Taylor et al., 2012). Furthermore, these effects may be adjusted through anthropogenic activities such as land-use change. Many recent studies devoted to evaluating the impact of climate change on groundwater availability (Woldeamlak et al., 2007; Maxwell and Kollet, 2008; Van Roosmalen et al., 2009; Jackson et al., 2011; Stisen et al., 2011; Taylor et al., 2012; Engdahl and Maxwell, 2015; Goderniaux et al., 2015; Havril et al., 2017). These studies often use coupled climate-land-surface-subsurface models to investigate the potential response of groundwater storages to the outer forcings under different climate scenarios. Compared with the near land-surface fluxes/storages (e.g., soil moisture, evapotranspiration), the groundwater reservoir is less vulnerable to extreme events (Maxwell and Kollet, 2008). The slow response of groundwater to climate variability can be explained by the highly dynamic surface water/groundwater interaction, the existence of a variably thick unsaturated zone, and the big volume of groundwater storage. Quantification of uncertainty in future water resource projections and travel times (decades to centuries) of the regional groundwater system is critically important for regional water sustainability.

Due to the diverse patterns of the terrestrial water cycle in regions under different climate conditions, climate change will have diverse impacts on the groundwater recharge change. Sandström (1995), for instance, found that in Tanzania, a 15% decline in precipitation, without any change in air temperature, will result in a 40-50% decline of groundwater recharge, indicating a potential amplified change of recharge compared to that of precipitation. While some studies found an increasing trend of recharge in some regions (Brouyère et al., 2004; Van Roosmalen et al., 2009), others indicate that climate change will likely lead to decreased recharge rates (Pulido-Velazquez et al., 2015; Woldeamlak et al., 2007; Havril et al., 2017). The changes of recharge, regardless of the sign of change, will significantly influence the groundwater levels and may lead to ecological problems such as the vanishing of wetlands (Havril et al., 2017). Modification of groundwater recharge will control the flow paths and travel times of pollutants, which is critical to the sustainability of the regional groundwater system. Moreover, modification of groundwater recharge can change the age distribution for water in both the vadose zone and the saturated zone, as well as significantly change the composite age distribution (Engdahl and Maxwell, 2015).

Groundwater travel time distribution (TTD) is a robust description of the storage and transport dynamics within aquifers under various external forcings. It has many implications for hydrogeological and environmental studies. For instance, significant time-lags of the streamflow response to external forcings have been observed by multiple studies (Howden et al., 2010; Stewart et al., 2012; Jing et al., 2019). Besides, the legacy pollutants in groundwater reservoirs can have a great impact on the total pollutant loads for agricultural catchments (Wang et al., 2016; Van Meter et al., 2017). Groundwater TTD, as a lumped description of the heterogeneous aquifers, sheds light on the assessment of groundwater responses to non-point source contamination subjected to a changing climate and land use (Böhlke, 2002; Engdahl and Maxwell, 2015).

Although there are plenty of studies that have focused on assessing the impact of future climate change on groundwater recharge (Tillman et al., 2016; Crosbie et al., 2013; Jyrkama and Sykes, 2007; Pulido-Velazquez et al., 2015), groundwater budget (Pulido-Velazquez et al., 2015; Engdahl and Maxwell, 2015; Havril et al., 2017), and groundwater-surface water exchange (Scibek et al., 2007; Smerdon et al., 2007), there is an absence of a systematic evaluation of both the groundwater quantity and TTDs under different warming levels that incorporates the uncertainties in both climate projections and hydrological parameterizations. In this study, we analyze the response of groundwater (quantity and TTDs) to the 1.5, 2, and 3 °C global warming levels (above the preindustrial levels) in a central German basin (Nägelstedt) using a coupled hydrological model mHM-OGS (Jing et al., 2018). The key questions we aim to answer are: (1) How can the flow and transport conditions of a regional groundwater system in future decades differ from the historical period under different warming levels? (2) Can we quantify the degree of different uncertainty sources (e.g., uncertainties in climate projections and groundwater models) and their influences on the resulting groundwater simulations? To answer these questions, we pay particular attention to the assessment of the long-term effect of climate change on the regional groundwater systems considering the buffering effect of groundwater aquifers.

This paper is organized into several sections: Section 2 describes the basic topographical, geometrical, and geological properties of the study area. Section 3 introduces the methodology and materials for this study. Section 4 shows the setup and validation for the mHM and OGS models. The simulation results are presented in Section 5. A comprehensive discussion on the simulation results is displayed in Section 6, and the main conclusions are drawn at the end of this section.

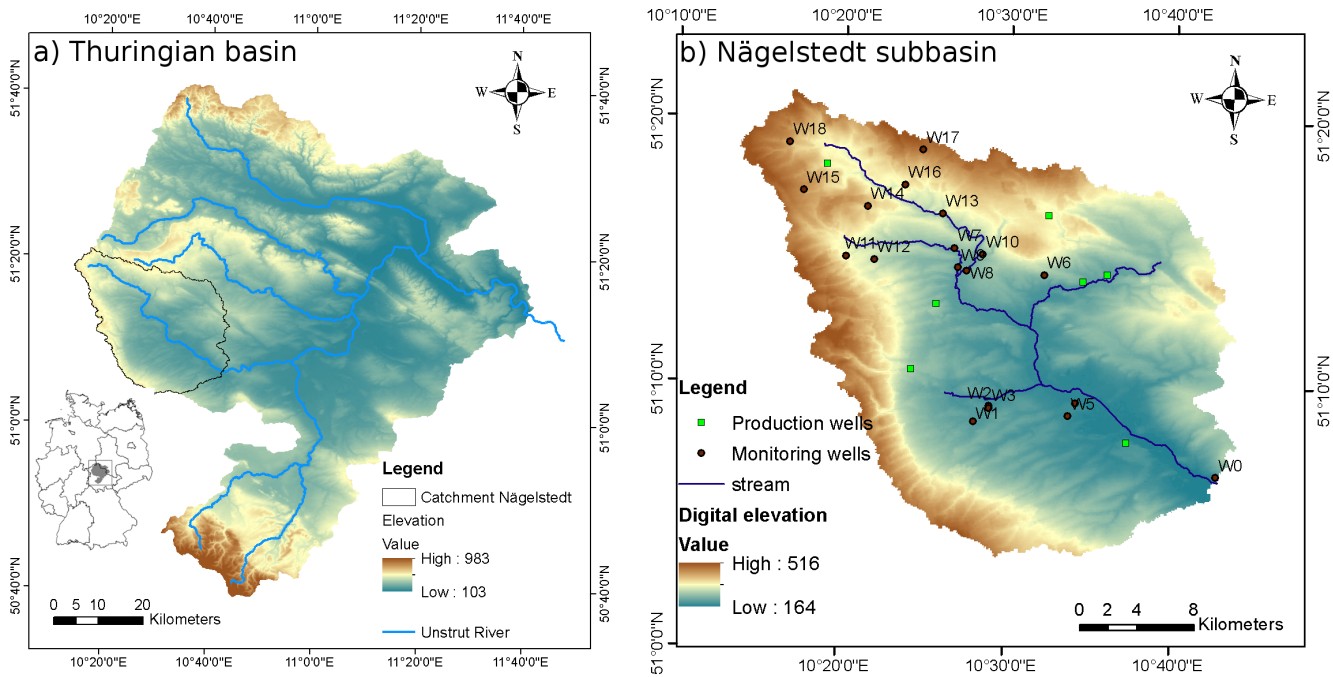

**Figure 1.** Study area and locations of pumping and monitoring wells within the Nägelstedt basin. Panel a) shows the relative position of Nägelstedt basin in the Thuringian Basin and Panel b) shows the locations of pumping and monitoring wells in Nägelstedt basin.

## 2  Study Area

As a sub-basin of the Thuringian basin, the Nägelstedt basin is located in central Germany and it has an area of about 850 km$^2$ (Figure 1). It is a headwater catchment of the Unstrut river. The Unstrut river is a typical, meandering lowland river with only moderate flow velocity under natural conditions. The mountains surrounding the Nägelstedt basin drain almost simultaneously

5    into the Unstrut during heavy precipitation events, which in the past led to regular, prolonged flooding of large parts of the floodplains. The topographic elevations of this catchment range from 164 m at the southeastern lowland to 516 m in the Hainich mountainous region. This region is classified as a Cfb climate region based on the Köppen-Geiger classification, where Cfb stands for warm temperate, fully humid, and warm summer climate (Kottek et al., 2006). It shows a leeward decreasing trend of areal precipitation and rising mean air temperature from the eastern Hainich ridge to the Unstrut Valley (Kohlhepp et al.,

10   2017). In the larger Thuringian basin, groundwater has been intensively extracted for domestic, industrial and agricultural uses. About 70% of the freshwater requirement for Thuringia is satisfied by groundwater (Wechsung et al., 2008).

   The extremely fertile soils in the meadows (wet black soil and loess) make the Thuringian basin one of the best agricultural basins in Germany. Approximately 88% of the total land use of this region is regarded as arable land (Wechsung et al., 2008). At the same time, the proportion of woodland and grassland has fallen sharply, leading to an extreme reduction in biodiversity

15   in these areas (Wechsung et al., 2008).

The stratigraphy in this area is characterized by a succession of carbonate–siliciclastic alternations. The main aquifer system consists of several sedimentary rocks, including the Middle Keuper (km), the Lower Keuper (ku), the Upper Muschelkalk (mo), the Middle Muschelkalk (mm), and the Lower Muschelkalk (mu) (Seidel, 2003). The Middle Keuper consists of a marly series with gypsum and dolomite, whereas the Lower Keuper is constituted of grey clays and dolomitic limestone. The Upper Muschelkalk (Hauptmuschelkalk) is mainly made up of shelly limestone, marl, and dolostone. The Middle Muschelkalk consists mainly of evaporites (gypsum, anhydrite, and halite), meanwhile, the Lower Muschelkalk consists of limestone and marls (Seidel, 2003; McCann, 2008; Jochen et al., 2014). Karstification occurs in the Muschelkalk formation, but has proved to be limited or concentrated in specific zones in this area (Kohlhepp et al., 2017).

The Nägelstedt basin is chosen as the study area for the following reasons: (1) It is a typical agricultural basin where potential non-point source contamination may threaten the sustainability and resilience of groundwater, and (2) the critical zone (CZ) in Nägelstedt basin has been comprehensively investigated using infrastructure platform from the Collaborative Research Center AquaDiva (Küsel et al., 2016; Kohlhepp et al., 2017).

## 3 Methodology and Materials

To investigate the impact of different climate change scenarios, we modified the modeling framework from EDgE (EDgE – End-to-end Demonstrator for improved decision-making in the water sector in Europe) and HOKLIM (HOKLIM – High-resolution Climate Indicators for 1.5 Degree Global Warming) projects through coupling it to a three-dimensional subsurface model (Thober et al., 2018; Marx et al., 2018; Samaniego et al., 2018). Specifically, we use temperature and precipitation derived from five GCMs under three different RCPs to force the mesoscale Hydrologic Model (mHM), aiming to derive the land surface fluxes and states under different future warming scenarios. The projected recharges from mHM calculations are fed to the groundwater model OpenGeoSys (OGS) for the assessment of groundwater quantity and TTDs.

### 3.1 Climate scenarios

We use five General Circulation Models (GFDL-ESM2M, HadGEM2-ES, IPSL-CM5A-LR, MIROC-ESMCHEM, and NorESM1-M) gathered from the Coupled Model Inter-comparison Projects 5 (CMIP5) to provide the climate variables to the mHM model. Temperature and precipitation are derived from these GCMs under three representative concentration pathways (RCPs; RCP2.6, RCP6.0, and RCP8.5), which are available from the ISI-MIP project (Warszawski et al., 2014). RCPs are representations of emission scenarios, with RCP2.6, RCP6.0, and RCP8.0 representing low, medium, and high emission scenarios, respectively. This multimodel ensemble approach enables the consideration of uncertainty in climate modeling. Climate variables from GCMs are downscaled to a 0.5 ° spatial resolution employing a trend-preserving bias correction approach (Hempel et al., 2013). The trend-preserving bias correction approach is capable of representing the long-term mean and extremes of catchment state variables (Hempel et al., 2013). The 0.5-degree data is further interpolated onto $5 \times 5$ km$^2$ grids employing a external drift kriging (EDK) approach. The EDK approach can incorporate altitude effects at the sub-grid scale and has been successfully used in many studies (Zink et al., 2017; Thober et al., 2018; Samaniego et al., 0).

**Table 1.** Time periods of 1.5, 2, and 3 °C global warming in five GCMs under three RCPs.

| Warming level | RCPs | GFDL-ESM2M | HadGEM2-ES | IPSL-CM5A-LR | MIROC-ESM-CHEM | NorESM1-M |
|---|---|---|---|---|---|---|
| | 2.6 | - | 2007–2036 | 2008–2037 | 2006–2035 | 2047–2076 |
| 1.5 °C | 6.0 | 2040–2069 | 2011–2040 | 2009–2038 | 2012–2041 | 2031–2060 |
| | 8.5 | 2021–2050 | 2004–2033 | 2006–2035 | 2006–2035 | 2016–2045 |
| | 2.6 | - | 2029–2058 | 2060–2089 | 2023–2052 | - |
| 2 °C | 6.0 | 2060–2089 | 2026–2055 | 2028–2057 | 2028–2057 | 2054–2083 |
| | 8.5 | 2038–2067 | 2016–2045 | 2018–2047 | 2017–2046 | 2031–2060 |
| | 2.6 | - | - | - | - | - |
| 3 °C | 6.0 | - | 2056–2085 | 2066–2095 | 2055–2084 | - |
| | 8.5 | 2067–2096 | 2035–2064 | 2038–2067 | 2037–2066 | 2057–2086 |

We use the period 1971–2000 to represent current climate conditions because 1991–2000 is the latest decade that is available in the GCM data. The GCM data from this period serves as a baseline scenario for the future projection of climate change. A time-sampling approach is applied to estimate the period for different global warming levels of 1.5, 2, and 3 °C (James et al., 2017). The five GCMs have different degrees of climate sensitivity due to the different climate projections, therefore providing
different meteorological forcings to the mHM model. Specifically, different periods of 1.5, 2, and 3 °C global warming are estimated by five GCMs under three RCPs (Table 1). The GCM data are divided into several 30-year periods. The first 30-year period that ever surpasses a certain warming level (1.5, 2, or 3 °C) is defined as the period of this warming level. We note that some combinations of GCMs and RCPs cannot be identified for the future climate projection before 2099, resulting in a total of 35 GCM/RCP combinations being used in this study (Table 1).

**3.2   The mesoscale Hydrologic Model (mHM)**

The disaggregated meteorological data are used as meteorological forcings of the mesoscale Hydrological Model (mHM) for a daily simulation. mHM is a spatially explicit distributed hydrologic model that applies grid cells as primary hydrologic units and accounts for multiple hydrological processes including infiltration, surface runoff, evapotranspiration (ET), soil moisture dynamics, snow accumulation and melting, groundwater recharge, and discharge generation. mHM is forced by hourly or
daily meteorological forcings (e.g., precipitation and temperature), and uses accessible physical characteristics including soil textural, vegetation, and geological properties to estimate the spatial variability of parameters utilizing its unique Multiscale Parameter Regionalization (MPR) technique (Samaniego et al., 2010; Kumar et al., 2013). The MPR technique is capable of coping with fine-scale features because the effective model parameters are regionalized based on the underlining subgrid-scale information using a consistent upscaling algorithm. The mHM simulations have been successfully established across Europe,
and the simulated land surface fluxes have been verified by eddy-covariance stations across Germany (Zink et al., 2017).

### 3.3 OpenGeoSys (OGS)

The porous media simulator OpenGeoSys (OGS) is used to simulate regional groundwater flow and transport processes. There are two OGS versions available – OGS-5 and OGS-6, and we use OGS-5 exclusively in this study. OGS has been successfully coupled to mHM through a coupling interface – mHM-OGS (Jing et al., 2018). The coupling interface interpolates the grid-based recharge produced by mHM into the nodal recharge values spreading over the top surface of the OGS-mesh. In doing so, mHM and OGS are dynamically coupled as a surface-subsurface model such that the potential recharge produced by mHM can be fed to OGS and serves as the outer forcing of the groundwater module (Jing et al., 2018). Specifically in this study, we feed the projected $5 \times 5$ km$^2$ recharge from mHM under future climate scenarios to the coupling interface (mHM-OGS) to run the groundwater model. OGS is based on the finite element method (FEM) and solves the partial differential equations (PDEs) of fluid flow employing linear/non-linear numerical solver. OGS is capable of simulating single processes including saturated zone flow, unsaturated zone flow, and solute transport, as well as coupled processes including saturated/unsaturated flow, multi-phase flow, and reactive transport. Specifically in this study, OGS is used to compute three-dimensional saturated zone flow.

Moreover, a Lagrangian particle tracking method – namely random walk particle tracking (RWPT) – is used to track flow pathways and compute TTDs of water parcels (Park et al., 2008a, b; Jing et al., 2019). The RWPT method assumes that the advection process is deterministic, while the diffusion/dispersion processes are modeled stochastically (Park et al., 2008a). The RWPT method has been widely used to account for reactive transport processes and travel times (Park et al., 2008b; Jing et al., 2019; Engdahl, 2017).

### 4 Model setup

We designed two parallel numerical experiments to investigate the effect of uncertainties in both the climate and groundwater models on the groundwater resources. For the evaluation of climate uncertainty, 35 GCM/RCP pairs are used, whereas one parameter set (related to hydrogeological features) is used for the groundwater model. In parallel, to assess the parameter uncertainty in the groundwater model, one sole climate realization is used, whereas many realizations of hydraulic conductivity fields constrained by the observations are used for the groundwater model. Specifically, this climate realization is the ensemble average of all 35 members of GCM/RCP combinations.

### 4.1 mHM model setup

The down-scaled meteorological dataset, corresponding to 5 GCMs, with a spatial resolution of $5 \times 5$ km$^2$ is used as the outer forcing of mHM. The model is set up across Europe and is forced with spatially distributed meteorologic observations obtained from the E-OBS dataset (Haylock et al., 2008; Samaniego et al., 0). Global parameters of mHM are calibrated against discharge observations from the GRDC database. All ensemble simulations are established with the same morphological, land use, and soil type data to keep the relevant parameters consistent throughout this study. Furthermore, the mHM model was validated

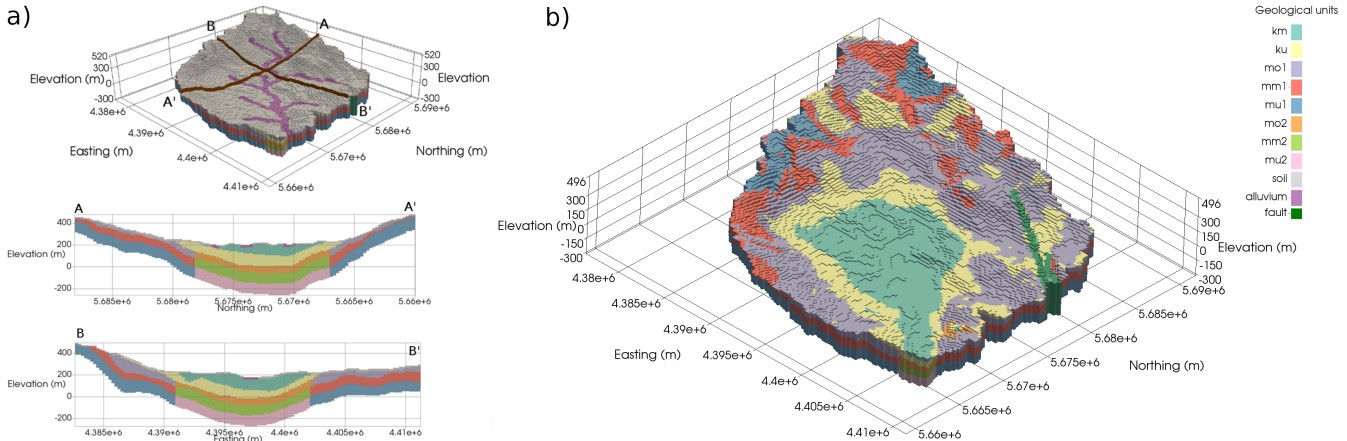

**Figure 2.** Geological zonation and three-dimensional mesh for the aquifer system in Nägelstedt basin (Jing et al., 2018). Panel (a) underlines the the spatial pattern of alluvium and soil layers. Panel (b) further displays the zonation of deep geological units. Full names of legends are listed as follows: km – Middle Keuper, ku – Lower Keuper, mo – Upper Muschelkalk, mm – Middle Muschelkalk, mu – Lower Muschelkalk.

using observations from many gauging stations across Europe with a period 1966-1995 (Marx et al., 2018; Thober et al., 2019; Samaniego et al., 0). The calibration-constrained parameter set is used for groundwater recharge projection. The projected groundwater recharge, with a spatial resolution of $5 \times 5$ km$^2$, is further downscaled to a $250 \times 250$ m$^2$ spatial resolution using the bilinear interpolation for establishing the fine-scale OGS groundwater model.

## 4.2 OGS model setup

A 25-m Digital Elevation Model (DEM) is used to determine the outer bounds of the catchment and the top surface elevation of the three-dimensional model domain. A three-dimensional stratigraphic mesh is set up based on the above information and bore log data from the Thuringian State Office for the Environment and Geology (TLUG) (Fischer et al., 2015). The mesh consists of 293,041 structured hexagonal elements with a size of 250 m in the $x$ and $y$ direction as well as with a 10 m resolution in the $z$ direction. The parameter zonation approach is used to represent the heterogeneity of hydraulic properties–hydraulic conductivity in this study. The geological zones within the three-dimensional mesh representing Nägelstedt catchment are displayed in Figure 2. Ten different sediment units are delineated based on the stratigraphy in this area, including Middle Keuper (km), Lower Keuper (ku), Upper Muschelkalk 1 (mo1), Upper Muschelkalk 2 (mo2), Middle Muschelkalk 1 (mm1), Middle Muschelkalk 2 (mm2), Lower Muschelkalk 1 (mu1), Lower Muschelkalk 2 (mu2), alluvium, and the uppermost soil layer (Figure 2). The geological unit "alluvium" represents sandy outwash and gravel near streams, whereas "soil" denotes the uppermost soil layer with a depth of 10 m.

For the uncertainty study of climate scenarios, a post-calibration hydraulic conductivity field sampled from many realizations that are all constrained by head observations is adopted for the OGS groundwater model (Table 2). In parallel, to assess the groundwater model uncertainty, 80 realizations of hydraulic conductivity fields randomly sampled from many hydraulic

**Table 2.** Hydraulic parameters used for ensemble simulations with different climate scenarios.

| Geological units | Hydraulic conductivity (m/s) | Porosity (-) |
|---|---|---|
| Middle Keuper (km) | $1.145 \times 10^{-4}$ | 0.2 |
| Lower Keuper (ku) | $3.714 \times 10^{-6}$ | 0.2 |
| Upper Muschelkalk 1 (mo1) | $3.936 \times 10^{-4}$ | 0.2 |
| Middle Muschelkalk 1 (mm1) | $2.184 \times 10^{-4}$ | 0.2 |
| Lower Muschelkalk 1 (mu1) | $2.258 \times 10^{-5}$ | 0.2 |
| Upper Muschelkalk 2 (mo2) | $3.936 \times 10^{-5}$ | 0.2 |
| Middle Muschelkalk 2 (mm2) | $2.184 \times 10^{-5}$ | 0.2 |
| Lower Muschelkalk 2 (mu2) | $2.258 \times 10^{-6}$ | 0.2 |
| alluvium | $1.445 \times 10^{-3}$ | 0.2 |
| soil | $3.026 \times 10^{-4}$ | 0.2 |

conductivity fields are used to cover a plausible range of values (Jing et al., 2019). Meanwhile, a uniform porosity of 0.2 is assigned to each geological layer (Table 2).

Given that this study is designed to assess the potential response of the regional groundwater system to global warming scenarios, a steady-state groundwater system could be assumed. This assumption is made because the future warming level is a long-term average, and on such a temporal scale (over the 30-year baseline), the short-time fluctuations of climate forcings are essentially damped in the regional groundwater system (Maxwell and Kollet, 2008).

The bottom and outer boundaries of the model domain are impermeable, and no-flow boundary conditions are assigned onto these geometries. The spatially distributed recharges estimated by mHM under future climate scenarios are mapped onto each grid node of the mesh surface by the model interface (mHM-OGS). Long-term averaged pumping rates are assigned as Neumann boundaries to each production well, wherein the pumping rates are obtained from the literature based on long-term historical data (Wechsung et al., 2008). The total long-term averaged pumping rate over the Nägelstedt catchment is 18 870 m$^3$/day, and it is set constant for all climate scenarios. A fixed head boundary is assigned to the main perennial streams including one mainstream and three tributaries (Figure 1). For the Lagrangian particle tracking model, about 100 000 spatially distributed particle tracers are injected through the top surface of the mesh. The spatial distribution of particle tracers follows the pattern of simulated diffuse recharges for each climate scenario.

### 4.3 Model calibration

We use the observed discharge and groundwater head over 50 years (1955-2005) to calibrate the mHM and OGS model. The established mHM model for the study area has been calibrated using the observed discharges at the outlet of the catchment in the previous study (Jing et al., 2018). The OGS groundwater model has also been successfully calibrated using the long-term averaged head observations at many monitoring wells (Figure 3). Figure 3 reveals that all 80 sets of hydraulic conductivity

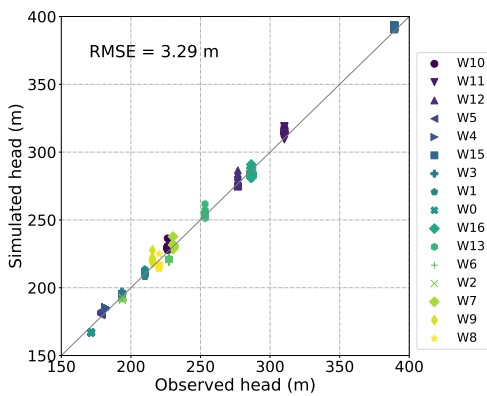

**Figure 3.** Groundwater model calibration: comparison of simulated to observed groundwater heads at several monitoring wells (W0 - W16) located across the study area (Figure 1) using 80 different hydraulic conductivity fields.

fields are compatible with the groundwater head observations with a small value of Root Mean Square Error (RMSE) being observed.

## 5  Results

In this section, we present the ensemble of simulated changes in groundwater recharges, levels and TTDs. For the sake of clarity, we use the plus sign to represent simulated values of increases and minus sign to represent decreases.

### 5.1  Climate impact on groundwater recharge

Relative changes of simulated mean annual recharge under 1.5, 2, and 3 °C warming for every GCM are shown in Figure 4. Projected changes of mean annual recharge vary from -4% to +15% for 1.5 °C warming level, and from -3% to +19% for the 2 °C warming level. The simulated changes under the 3 °C warming scenario range from -8% to +27%. The simulation results from 29 out of 35 total GCM/RCP combinations suggest an increase of groundwater recharge, while only 6 individual simulations projected decreased recharge rates. The projected changes are more dependent on the used GCMs than the RCPs, which can be expected because differences among RCPs are moderated by analyzing different warming levels. Under the same rising temperature and GCM, projected recharges still vary for different RCPs. The scale and distribution of precipitation change respond not only to temperature rise but also to employed RCPs (Mitchell et al., 2016; Thober et al., 2018). This is introduced by non-$CO_2$ forcing and the dependence of precipitation sensitivity to emission scenarios (Mitchell et al., 2016). This phenomenon indicates the necessity of considering multiple GCM/RCP combinations for providing a plausible range of predictive uncertainty. The ensemble averages of relative changes suggest an increase of 8.0%, 8.9%, and 7.2% for the 1.5, 2, and 3 °C warming, respectively. Meanwhile, the standard deviations (SDs) increase with the warming level. With the increase of the global warming level, the predictive variability in groundwater recharge is also expected to increase (Figure 4).

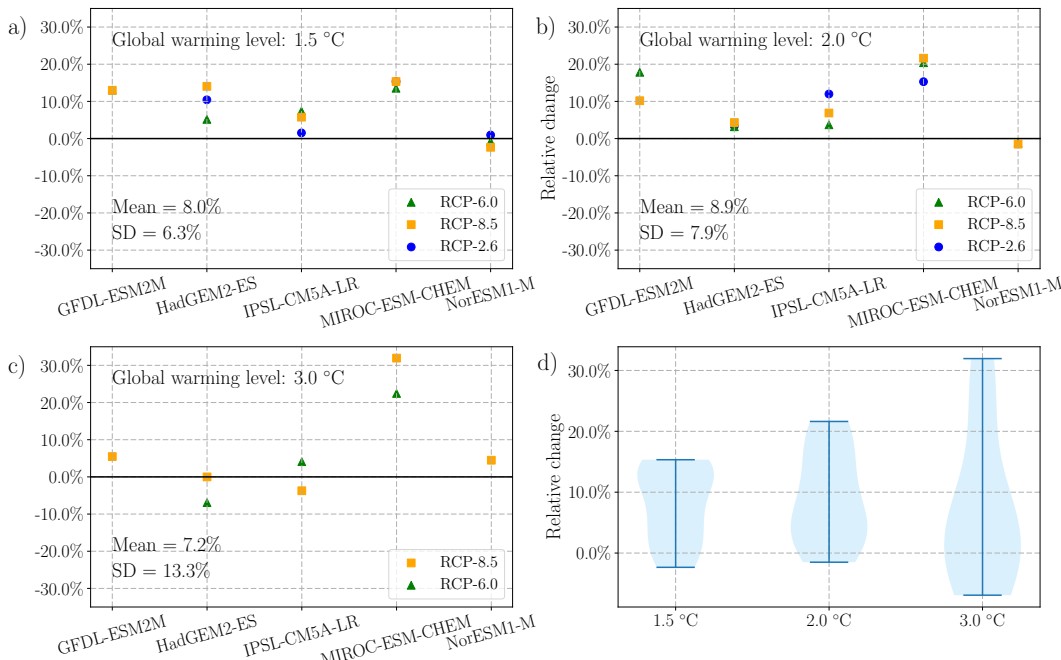

**Figure 4.** Projected changes in groundwater recharge rate under three warming scenarios compared to the baseline scenario 1971-2000. Panel a), b), and c) are the scatter plots showing the individual simulation results, and panel d) is the violin plot showing the uncertainty of ensemble simulations.

Generally, results indicate that the projected groundwater recharge rate is expected to be greater than the 1971-2000 average. The increases in groundwater recharge are below 20% in magnitude in the majority of GCM/RCP realizations, whereas three GCM/RCP realizations suggest a decrease of groundwater recharge in the study area. The simulation under 3 °C warming scenario represents the largest standard deviation, i.e., the highest uncertainty. Note that this uncertainty among different simulations is mainly introduced by the climate projection using various GCM/RCP combinations, given that mHM is the only hydrologic model used in this study and the underlying model parameterizations are the same for all simulations.

## 5.2 Climate impact on groundwater levels

Changes of simulated spatially distributed groundwater levels under future climate scenarios using the minimum, median, and maximum projected recharge are shown in Figure 5. Generally, the areas of topographically-driven flow (e.g., slope) appear to be more sensitive to the changes of recharge compared to the lowland plain. Under 1.5 °C warming scenario, the simulated groundwater levels using maximum recharges present an increase ranging from 0 to 10 m compared to those under the baseline scenario, whereas those using minimum recharges exhibit a slight decrease. Under the 2 °C warming scenario, groundwater levels are expected to increase compared to the base case using median and maximum projected recharges, whereas marginal differences can be found in the simulated levels using a minimum projected recharge. Under the 3 °C warming level, the

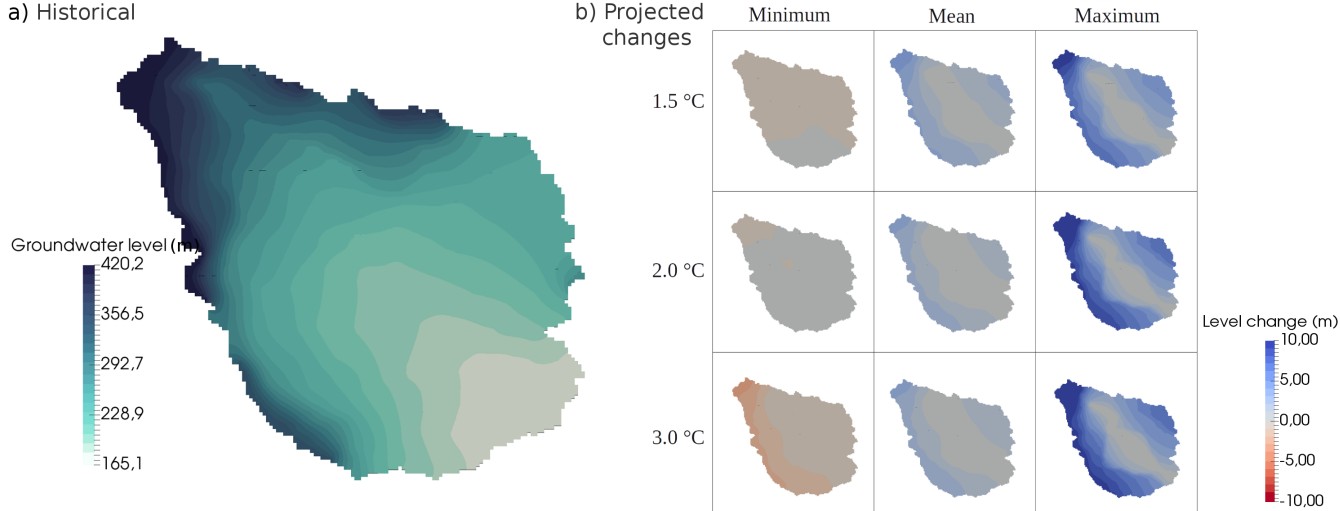

**Figure 5.** Contour maps of groundwater levels in Nägelstedt catchment. Panel a) shows the long-term average of groundwater levels a.b.m.l. (above mean sea level) in the historical period 1971-2000. Panel b) shows the changes in simulated groundwater levels under 1.5 °C, 2 °C, and 3 °C warming scenarios compared to the baseline scenario 1971-2000 using the maximum, median, and minimum projected recharges.

simulated changes in groundwater levels show the highest variation among the three warming levels. Simulations using the maximum recharge suggest a significant increase of groundwater level compared to the 1971-2000 historical average, while simulation using the minimum recharge results in a moderate decrease of groundwater levels (up to a decrease of 5 m at the northeastern mountain).

Figure 6 further shows the changes in groundwater levels at several monitoring wells, of which the locations are displayed in Figure 1. In general, changes in groundwater levels are induced by the changes in groundwater recharge such that more groundwater recharge results in higher groundwater levels and vice versa. The uncertainty of groundwater level changes increases with the warming levels, which can be evidenced by an increasing standard deviation of simulated groundwater levels from 2.20 m for 1.5 °C warming to 4.70 m for 3 °C warming (Figure 6). The projected changes in groundwater levels present a widespread variation associated with the variability of GCMs. Simulated groundwater levels tend to have the largest increase under three global warming levels for the MIROC-ESM-CHEM model. In contrast, simulated groundwater levels based on the NorESM1-M model show minimal changes compared to the baseline scenario. Estimated groundwater levels also respond to different RCPs, which is attributed to the RCP-dependent precipitation and its modification on recharge. Alternatively speaking, the uncertainty in RCPs is ultimately propagated from the RCP-dependent precipitation to the simulated groundwater levels. Although differing in magnitude, the changes in groundwater levels for different wells show a consistent trend (either increasing or decreasing) under the same GCM/RCP realization. The simulations show no systematic relationship between the change in groundwater levels and the change in global warming level, but they do indicate an increased variability in ground-

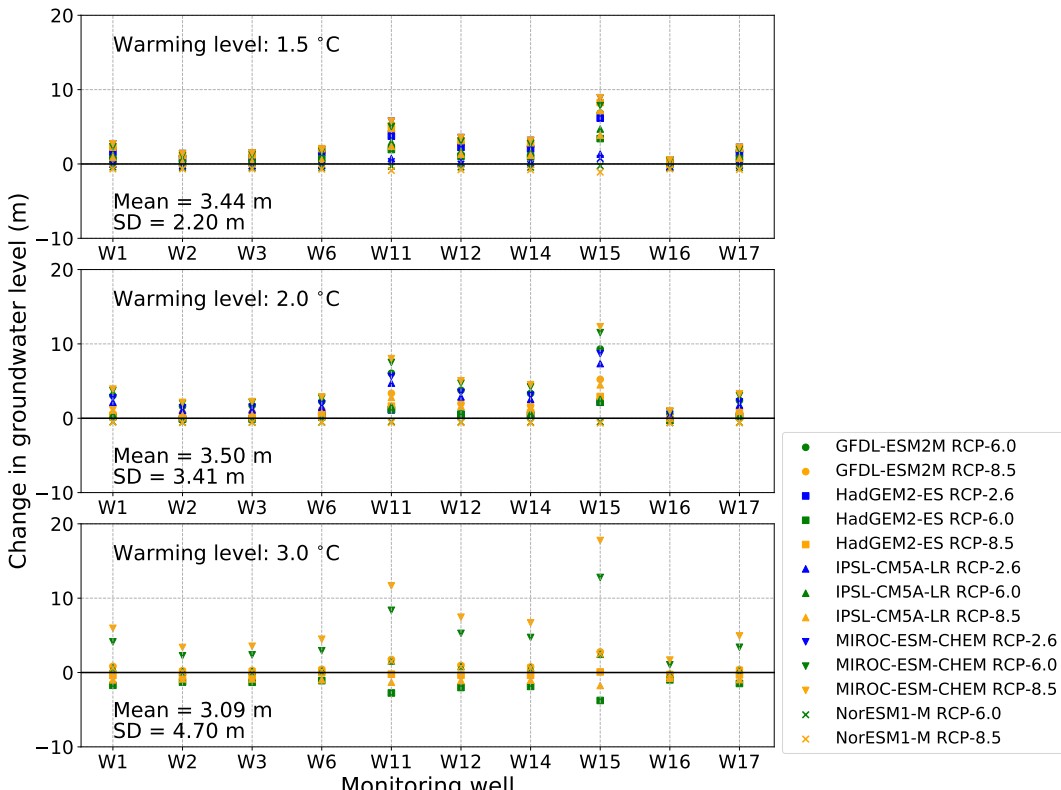

**Figure 6.** Changes of simulated groundwater levels in monitoring wells under three warming scenarios compared to the baseline scenario. The positions of monitoring wells are shown in Figure 1b.

water level change following the increased warming level – which can be evidenced by the increased standard deviation values from 1.5 to 3 °C warming level (Figure 6).

Overall, calculations of spatially distributed groundwater levels help to understand more of the response of groundwater quantity to the projected climate change, but they provide little clue on the change in the groundwater transport process.
5 Strong spatial variability in changes in groundwater levels reveals the high climate change sensitivity in mountainous areas and relatively low sensitivity in lowland plain areas.

### 5.3 Climate impact on groundwater travel time distributions (TTDs)

TTDs provide a robust description of the flow pathways of water parcels through the subsurface as well as the storage of groundwater within it. Simulated TTDs in Nägelstedt catchment under 1.5, 2, and 3 °C warming levels are shown in Figure 7.
10 Figure 7a shows the probability density function (PDF) of TTDs for the ensemble simulations. Generally, the simulated PDFs show a fairly consistent shape with a long tail extending to hundreds of years for all GCM/RCP combinations. The long-tail behavior of simulated TTDs can be explained by the direct influence of hydro-stratigraphic aquifer system, whereby some

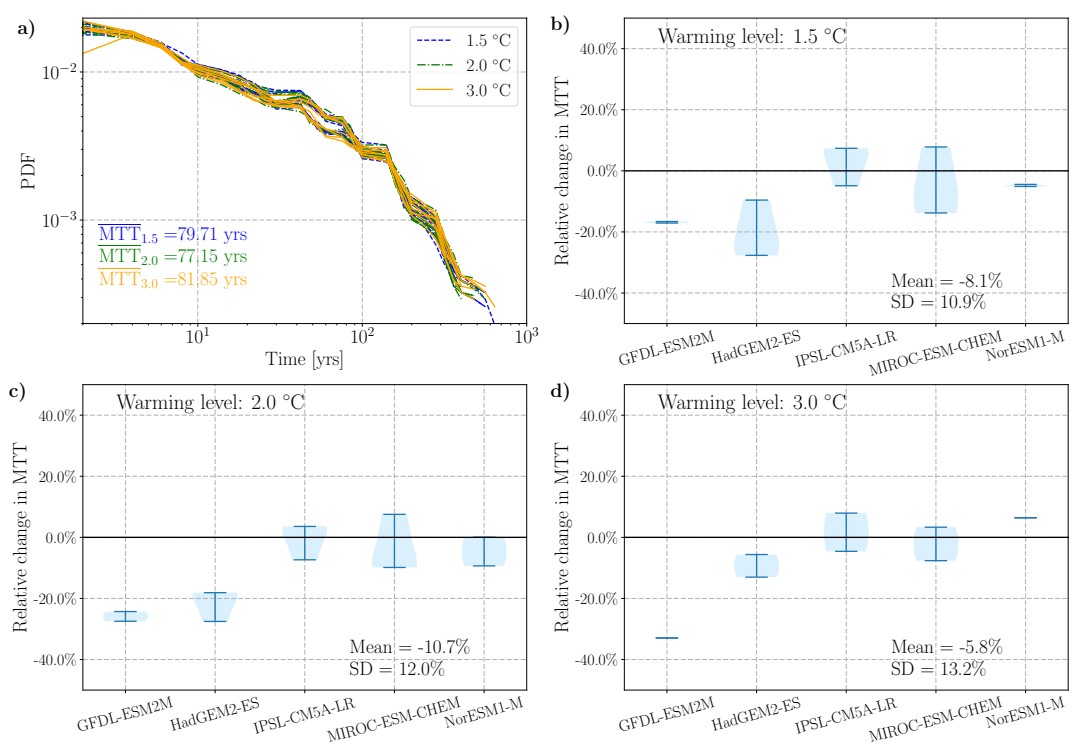

**Figure 7.** Simulated TTDs in Nägelstedt catchment under 1.5 °C, 2 °C, and 3 °C warming scenarios. Panel a) shows the probability density function (PDF) of TTDs for the ensemble simulations. Panels b), c), and d) show the relative changes of mean travel times (MTTs) under future climate scenarios compared to the base case.

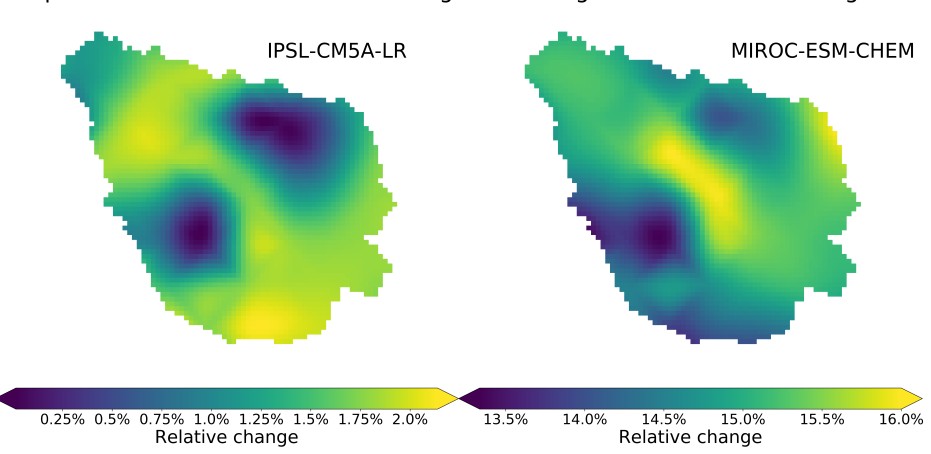

**Figure 8.** Spatial distributions of relative change in diffuse recharge using two different GCMs under 1.5 °C warming. This figure indicates varying spatial organizations of diffuse recharge change for different GCMs.

geological units present very low hydraulic conductivity values (e.g., mm2 and mu2) and therefore, remarkably slow down the movements of particles in these layers. The mean travel time (MTT), which by definition is the mass-weighted average of travel times for all water parcels within the simulated subsurface system, is a typical metric for characterizing the timescales of catchment storage. The calculated ensemble averages of MTTs for 1.5, 2, and 3 °C warming levels do not exhibit notable

differences (79.71, 77.15, and 81.85 years, respectively).

To analyze the changes of MTTs under the future climate scenarios, the relative changes of MTTs under 1.5, 2, and 3 °C warming levels are shown in panels b), c), and d) of Figure 7. In general, simulations using the data from GFDL-ESM2M and HadGEM2-ES tend to decrease MTTs compared to that of the baseline scenario. TTD Simulation results based on the IPSL-CM5A-LR and MIROC-ESM-CHEM, however, do not agree on the sign of changes in MTTs. A maximum relative

change of less than 10% is observed for these two model cases. The ensemble average shows that the MTT is expected to decrease in future periods, but a small number of ensemble simulations suggest an increase in MTT. This degree of variability is propagated from the variation in projected recharges corresponding to different GCM/RCP combinations. The simulations do not show any systematic relationship between the change in TTDs and the change in the warming level, but they do show an increased uncertainty in projected change in TTDs following the increased warming level – which can be demonstrated by

the corresponding increased standard deviation values of MTT from 1.5 to 3 °C warming (Figure 7).

Overall, changes in the simulated TTDs provide more overview on the groundwater system response to climate change and how the groundwater is impacted other than considering only the groundwater quantity. The simulated changes in MTTs exhibit a higher variability than the changes in groundwater levels. This is attributed to the fact that the simulated changes in recharge have varying details of spatial patterns for different GCM/RCP realizations (Figure 8). This spatial variability results

in a non-linear relationship between projected changes in MTT and those in the groundwater level. This observation is in line with the previous finding that TTDs are more sensitive to the spatial pattern of diffuse recharge than the groundwater levels are (Barthel and Banzhaf, 2015; Jing et al., 2019).

## 5.4  Predictive uncertainty related to the groundwater model

This subsection displays simulation results using 80 different hydraulic conductivity fields that are all conditioned by the

head observations and reality. The spread of hydraulic conductivity values in each geological unit can be found in Jing et al. (2019). Note that only one climate model is used for this group of simulations, which guarantees the simulation results are only controlled by different hydraulic conductivity values. Figure 9 displays the spread of changes in simulated groundwater levels and MTTs using 80 different hydraulic conductivity fields at 17 selected monitoring wells. The spread of results varies with the location of monitoring well, provided that the local topographic and hydraulic properties around each monitoring well

are different. Wells near the mainstream (e.g., W14 and W16) show smaller variations than those located far away from the mainstream (e.g., W1, W3, and W15), indicating the buffering effect of the groundwater aquifer. By comparing the spread of projected changes in groundwater levels in Figure 9 to those in Figure 6, we find that the spreads induced by different GCM/RCP are remarkably larger than those induced by accounting for different hydraulic conductivity fields. Moreover, the sign of projected changes in groundwater levels in Figure 6 can be either positive or negative, whereas those in Figure 9

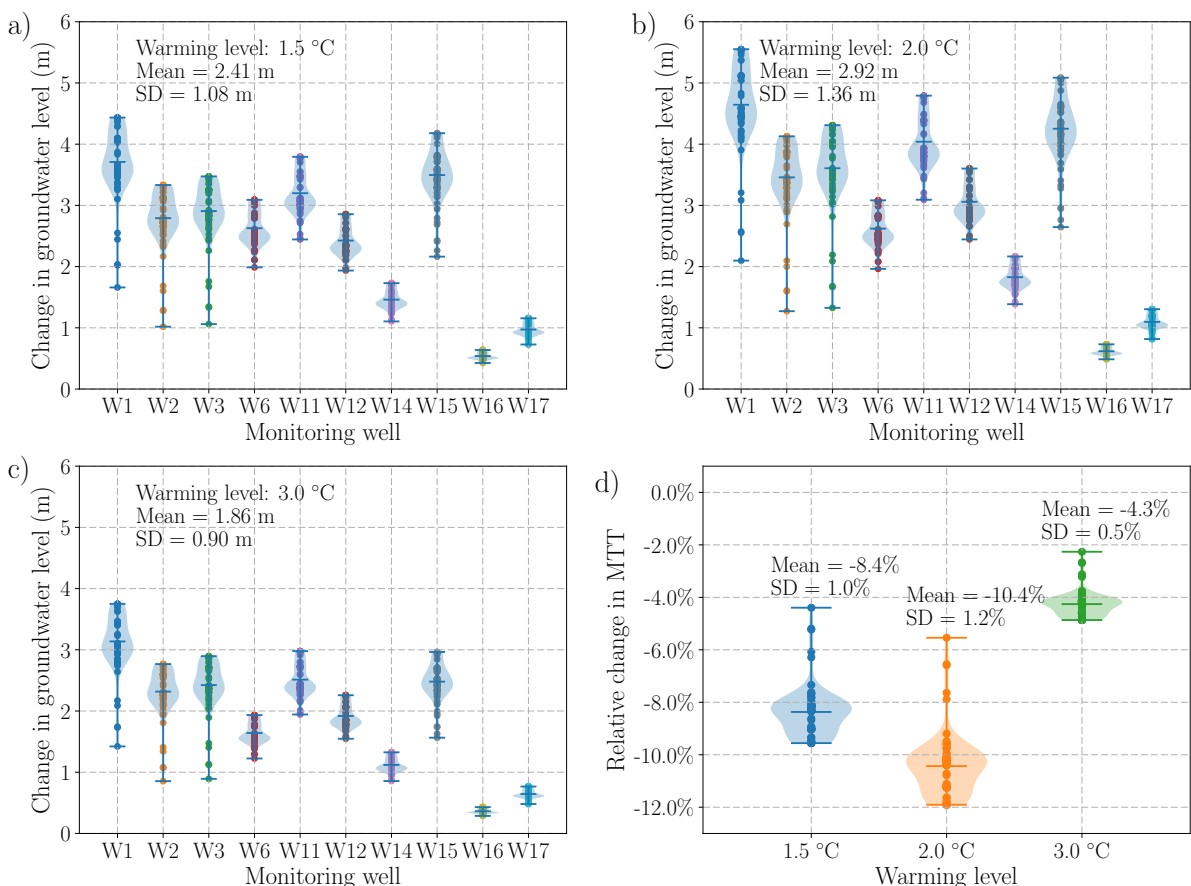

**Figure 9.** The predictive uncertainty in simulation results related to different hydraulic conductivity fields. Panel a), b), and c) show the changes in groundwater levels, whereas panel d) shows the projected relative changes in MTTs using 80 different hydraulic conductivity fields.

show a consistent positive sign. The spread of projected relative changes in MTTs ranges from -12.0% to -2.4%, which is also significantly smaller compared to that related to different climate models (Figure 9d). This comparison indicates that predictive uncertainties in groundwater level are primarily contributed by climate projections and secondly by hydraulic parameters.

## 6   Discussion and conclusions

5   In this study, we systematically explore the response of a regional groundwater flow system to different global warming scenarios by means of the sequentially coupled land surface model (mHM) and a groundwater model (OGS). The results of ensemble simulations manifest that groundwater recharge is likely to increase moderately under all three warming levels in a central German river basin. However, the ensemble simulations do not all agree on the sign of projected changes in groundwater

recharge. This is consistent with a previous finding that low flows are expected to increase slightly in this region under future climate scenarios , considering that baseflow is the main component of low flow and recharge feeds the baseflow(Marx et al., 2018). Similar increasing trend in climate-induced recharge rates has been suggested for other regions such as a northern European catchment (Treidel et al., 2012), high plains of USA (Cornaton, 2012), upper Colorado catchment (Tillman et al.,

2016) and Snake River basin (Sridhar et al., 2017).

The simulated changes in groundwater levels also manifest similar increases under all three warming levels, but show a strong spatial variability depending on the local topography and elevation. These changes can be critical to groundwater/surface water interaction because the increase or decrease in groundwater table would modify the dynamics of groundwater discharge into streams (Havril et al., 2017). In areas with complex topography and dense drainage network, rising groundwater level may

activate shallow groundwater flow paths and intensify shallow local flow pathways (Toth, 1963; Havril et al., 2017; Kaandorp et al., 2018). This way, the mixing behavior of groundwater storage can also change, because the activation of shallow flow paths will lead to a stronger systematic preference for discharging young water (Kaandorp et al., 2018; Jing et al., 2019). Moreover, changes of groundwater levels will impact the land surface processes such as evapotranspiration, soil moisture, and overland flow (Kollet and Maxwell, 2008; Huntington and Niswonger, 2012).

The influence of climate change on the catchment-scale groundwater TTDs is of critical importance to the sustainability of the groundwater system. Simulated MTTs suggest a moderate decline for three warming levels, which is not surprising since the travel time of water parcels is directly controlled by the recharge rate. A critical finding of this study is the non-linear relationship between the change in MTT and that in the groundwater level, which is mainly attributed to the spatial variability in diffuse recharge change. With weighted-average MTTs being at a centurial time scale, climate-induced variations

can significantly affect the long-term sustainability of the regional groundwater system (Engdahl and Maxwell, 2015). The projected decrease in groundwater MTTs will remarkably shorten the life span of non-point source pollutants (e.g., nutrient and pesticide) in groundwater aquifers and may introduce substantial changes in the spatiotemporal distributions of pollutant concentrations within the aquifer system. Given that the nutrient budget of the connected surface water body is linked with the groundwater system, the water quality of the surface water body in this region (e.g., the Unstrut river) will respond accordingly

in the future, although with a long delay (Molnat and Gascuel-Odoux, 2002; Böhlke and Denver, 1995). This observation is in line with many recent studies, wherein they highlight the importance of legacy nutrients in catchments as a reason for long-term catchment response (Haygarth et al., 2014; Van Meter et al., 2017).

One essential topic when assessing future climate impact is to quantify the uncertainties in the projected changes. This study provides original insights on the uncertainty propagation from the outer forcing (associated with climate models) and

the internal hydraulic properties (associated with groundwater models) to the groundwater travel times. Among simulations corresponding to different GCM/RCP combinations, simulated changes in hydrologic variables (e.g., recharge, groundwater level, and mean travel time) vary not only in their absolute values but also in sign (positive or negative) because of the large variations in different climate projections. The contribution of climate model induced to the predictive uncertainty is also found to be greater than that of hydraulic parameters in the groundwater model. Within the current modeling framework, predictive

uncertainty and error may also be introduced by other sources, such as the internal variability in climate projection using

different initial states, internal parameter uncertainty in the mHM model, and the down-scaling algorithm. Enhancements in climate projection and downscaling algorithms can effectively reduce the variability in the projected impacts of global warming on the regional groundwater system. Nevertheless, the dominant source of uncertainty is highly likely due to the climate projections of varying GCMs and RCPs (Taylor et al., 2012; Thober et al., 2018; Marx et al., 2018). Except for the above-mentioned climate projection uncertainty, other uncertainty sources have not been assessed in this study. This fact indicates that the range of predictive uncertainty quantified in this study is only a conservative estimate.

A potential limitation of the current modeling framework lies in the one-way coupling approach that do not account for the feedback from groundwater level change to the near-surface processes. The change in groundwater table can alter the partitioning of water balances, which further exerts a second-order impact on the groundwater level and travel times (Liang et al., 2003; Leung et al., 2011). A fully coupled system, based on a mixed form of the Richards equation to solve unsaturated and saturated zone flow simultaneously, is more realistic than the one-way coupled system. However, a fully coupled model consistently suffers from an expensive computational burden, which limits its applicability in large-scale real-world case study It also introduces extra parameters that are essentially unknown at the catchment scale. The current one-way coupling, although less accurate than the two-way coupling, is computationally more efficient – allowing us to understand the first-order control of climatic variability on groundwater characteristics (groundwater level and travel times). The applied Lagrangian particle tracking in the 3D groundwater model is computationally very expensive. The total computational time performing a single model run is around 14 days using 8 cores on a computer cluster facility. Moreover, we have successfully demonstrated the utility of this model for adequately capturing the observed behavior of groundwater levels across the study basin (Jing et al., 2018). Consequently, the one-way coupling method used here is a practical choice, allowing us to perform the large ensemble scenarios with reasonable computational resources (and time).

The second potential limitation of this study is the discrepancy between the fine-resolution groundwater model and coarse-resolution mHM simulations. mHM simulations in this study were established within the scope of the EDgE/HOKLIM project, which focuses on the impact of future climate scenarios on European water resources. All databases used for mHM model setup are on a European scale and typically have coarse spatial resolutions (e.g., $5 \times 5$ km$^2$). Although the MPR technique embedded in mHM facilitates the characterization of subgrid-scale features, it does not guarantee that all subgrid-scale features can be captured if the resolutions of input data are too coarse (Samaniego et al., 2010; Rakovec et al., 2016). We note that this is a common problem when utilizing coarse-resolution forcings to drive fine-resolution physically-based models. In this respect, simulation results in this study can be considered as first-order approximations based on currently available databases. The conclusions drawn in this study can be tentative and therefore, open to revision.

The steady-state nature of groundwater simulations used here is reasonable for the assessment of long-term climate impact on regional groundwater system because 1) it reduce the computational burden, 2) the temporal fluctuations under the future climate cannot be reasonably projected, and 3) high-frequency fluctuations in external forcings have minor influences on long-term TTDs (Engdahl, 2017). However, analyzing the likely future changes in transient behavior of groundwater dynamics can be important for many cases where the temporal scale is small and the input forcings are highly dynamic. In recent years, the

subject of the transient behavior of TTDs has become more and more prevalent in groundwater hydrology (Woldeamlak et al., 2007; Cornaton, 2012; Engdahl, 2017).

We note that the results in this study are only suitable for the Nägelstedt site in central Europe. In other regions of Europe, groundwater recharge change induced by global warming may have distinct behaviors than those shown in this study. For example, some studies indicate a decrease in groundwater quantity in Mediterranean regions due to the decrease in projected precipitation (Pulido-Velazquez et al., 2015; Moutahir et al., 2016). Besides, baseflow is also expected to decrease, leading to a potential increase in drought in Mediterranean regions (Marx et al., 2018; Samaniego et al., 2018).

We only consider the direct impact (i.e., impacts exerted through changed precipitations) of climate change on the regional groundwater system. Interactions between the climate and groundwater are exacerbated by land-use change, which is mainly exerted by the intensification of irrigated agriculture. In South Australia and the southwest U.S., the transition from natural catchments to rain-fed cropland significantly alters the groundwater storage through the increase in recharge (Taylor et al., 2012). These indirect influences of global warming on groundwater systems has not been considered in this study. Such influences can be a dominant factor threatening the local groundwater system for many regions worldwide (Wada et al., 2010; Taylor et al., 2012). Future investigations are needed to incorporate both the direct and indirect impacts of global changes on the sustainability of the regional groundwater system.

To summarize, climate change can significantly alter the quantity and travel time behavior of the regional groundwater system through the modification of recharge, especially at longer time scales. Ensemble simulations indicate remarkable uncertainties in projections of future regional groundwater quantity and travel times, which are introduced primarily by the driving climate projections, and secondly by hydrologic/groundwater model parameterizations. In the study domain, moderate absolute changes in recharge rates, groundwater levels, and travel times that are nonlinearly related to the varying level of global warming are found. However, the variability of these changes increases with the warming levels that might also affect the cost of managing the groundwater system. Therefore, it is still advisable to restrain global warming to 1.5 °C and avoid global warming of 3 °C.

*Code availability.* The coupled model mHM-OGS can be acquired via the following online repository: https://doi.org/10.5281/zenodo. 1248005. The mHM source code is available from: http://git.ufz.de/mhm/mhm. The OGS-5 source code is available from: https://www. opengeosys.org/ogs-5.

*Author contributions.* Conceptualization and methodology, M.J, R.K, F.H.; software, M.J., R.K., S.T, L.S, O.R.; validation, formal analysis and investigation, M.J.; resources, M.J., R.K., S.T, L.S; writing—original draft preparation, M.J.; writing—review and editing, S.T., F.H., R.K., O.R.; visualization, M.J.; supervision, S.A.

*Competing interests.* The authors declare that they have no conflict of interest.

*Acknowledgements.* This study was partially performed under a contract for the Copernicus Climate Change Service (edge.climate.copernicus.eu). ECMWF implements this service and the Copernicus Atmosphere Monitoring Service on behalf of the European Commission. This work is also supported by the Deutsche Forschungsgemeinschaft via Sonderforschungsbereich (CRC 1076 AquaDiva) and the German Ministry for

5 Education and Research within the scope of the HOKLIM project (www.ufz.de/hoklim; grant number 01LS1611A). We acknowledge Sabine Sattler from Thuringian State Office for the Environment and Geology (TLUG) for providing geological data. We also acknowledge people from various organizations and projects for the datasets that are used in this study, which include ISI-MIP, JRC, NASA, GRDC, BGR, and ISRIC. We acknowledge the Chinese Scholarship Council (CSC) for supporting Miao Jing's work.

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
