# Peer review of "Assessing the response of groundwater quantity and travel time distribution to 1.5, 2 and 3 degrees global warming in a mesoscale central German basin"

_Hydrology and Earth System Sciences, 2019_

## Referee Comment (RC1) · Anonymous Referee #1 · 5 Apr 2019

This manuscript analyzes the possible response of groundwater to climate change scenario obtained by combining a few Global and Regional Climate models under three warming levels (1.5, 2 and 3 oC). The analysis is conducted for the aquifer of a small catchment (850 kmˆ2) located in Germany with a total of 35 combinations of Global and Regional Climate Models. The resulting precipitation and temperature scenarios are used into a mesoscale hydrological model (mHM) to simulate vertical hydrological flow exchanges and the resulting infiltration is used as spatially variable recharge of a groundwater model (OpenGeoSys, OGS).

[Discussion paper]

[Figure]

The manuscript is well written and technically sound (see below for a few comments) and the topic is of interest, given the relevance of groundwater.

General comments

From the description provided in the methodology and materials section I argue that mHM and OGS, the mesoscale hydrological model and the groundwater model are decoupled. In fact, at line 13 of page 5 I read "The projected recharge from mHM calculations are fed to the groundwater model OpenGeoSys (OGS) for the assessment of groundwater quantity and travel time distribution." From this short description I concluded that vertical fluxes, including those of the unsaturated zone are modeled with mHM and that the resulting "deep" infiltration is used as recharge (i.e., as boundary condition) for OGS. In doing that the processes of infiltration and flow inside the aquifer are decoupled. This may be reasonable when the water table is deep, but I am wondering what is the impact of this assumption on the simulations showing significant rises of the water table. In this situation decoupling leads to significant errors as the water table gets close to the ground surface. This is relevant both for water levels and the following travel time analysis. The authors provide only limited information on this important point and do not discuss its implications in term of representativity of the model. Decoupling these two processes is a great advantage from a modeling point of view, but I am not sure it can be actually introduced at least in the scenarios showing the largest increase of groundwater levels.

Another aspect that is not fully explained is the validation of the groundwater simulations. The authors touch very briefly this point by saying that mHM has been validated at the European scale in a previous paper, but what about this specific small catchment or the larger, but still small compared to the European scale, Thuringian catchment? And what about the groundwater model? At page 8, line 17 I read: "The post-calibrated values of the hydraulic conductivity in each geological unit obtained from a previous study are assigned to the corresponding geological layers of the mesh (Jing et al., 2018a). Meanwhile, a uniform porosity of 0.2 is assigned to each geological layers

(Table 2)". In a previous work (Jing et al., 2018a) the authors presented a comparison between observed and simulated heads at a number of observation wells (Figure 5 and related text on section 3.2.5) and for a number of recharge scenarios. The analysis is based on 400 calibrated Monte Carlo realizations and I am wondering if the authors used all the 400 realizations in the present work, or just one, in the latter case what was the criteria used to assign the hydraulic conductivity? Figure 5 of the previous paper shows apparently a good reproduction of the observed heads, but what puzzled me is that the standard deviation of the error is 4.6 m, a rather large portion of the variation presented in this manuscript as an effect of climate change.

The third issue I would like to comment is uncertainty. My impression is that combining a large number of GCM-RCM pairs introduced a large variability of the meteorological forcing and therefore to water levels and travel time distributions, which should be validated in some way. According to the presented analysis it is very difficult, if not impossible, to sort out unrealistic scenarios, or weight less them in the ensemble. On the other hand, uncertainty in the hydrological models is neglected, in particular that related to the groundwater model.

Overall I feel that the manuscript, although interesting and pleasant to read, cannot be accepted for publication in the present form. As it is the manuscript reads like an application of previously published modeling efforts, with little analysis of the results. However, I see value in what the author presented and I think that with some extra effort they may accommodate the above drawbacks, by explaining more the underlying hypotheses and limitations of the current analysis and improving and enriching the discussion of simulation results. With these changes I think this manuscript will be a valuable contribution to the community.

Minor comments Page 9 line 6: The recharge seems to increase almost linearly with the temperature, which is strange considering the many nonlinearities involved in the infiltration process;
Page 9, line 9: "The projected....." This seems to suggest that the number of combinations of GCM and RCM can be reduced, or even that similar results can be obtained by using only the GCM. Please elaborate a bit more.

Page 14, line 17 and following: this sentence is vague. Have these shallow local flow pathways actually been observed in the simulations and how realistic are they?

Page 15 line, 25: this disclaimer, saying that uncertainty may be even larger, since some uncertainty sources have not been considered is somewhat alarming because it casts doubts on the interpretation of the results.

Page 16, line 20: How can be that first-order effects of climate change are small and second-order effects are not negligible? If for not negligible you mean that they are however smaller than first-order effects, what is the reliability of their estimate considering the large uncertainty affecting these simulations? Please elaborate more

---

## Author Comment (AC1) · 30 Apr 2019

**Reply to Referee Review 1**

We acknowledge the referee reviewer for those comprehensive and insightful comments. Our responses to the reviewers' comments are given below. The original comments from referee reviewer 1 were marked with blue color, and our response in black. The page and line numbers in our responses refer to those in the marked copy of the revised texts.

**General comments**

From the description provided in the methodology and materials section I argue that mHM and OGS, the mesoscale hydrological model and the groundwater model are decoupled. In fact, at line 13 of page 5 I read "The projected recharge from mHM calculations are fed to the groundwater model OpenGeoSys (OGS) for the assessment of groundwater quantity and travel time distribution." From this short description I concluded that vertical fluxes, including those of the unsaturated zone are modeled with mHM and that the resulting "deep" infiltration is used as recharge (i.e., as boundary condition) for OGS. In doing that the processes of infiltration and flow inside the aquifer are decoupled. This may be reasonable when the water table is deep, but I am wondering what is the impact of this assumption on the simulations showing significant rises of the water table. In this situation decoupling leads to significant errors as the water table gets close to the ground surface. This is relevant both for water levels and the following travel time analysis. The authors provide only limited information on this important point and do not discuss its implications in term of representativity of the model. Decoupling these two processes is a great advantage from a modeling point of view, but I am not sure it can be actually introduced at least in the scenarios showing the largest increase of groundwater levels.

Response:

We fully agree with the reviewer that groundwater dynamics can alter land surface response and feedback to the climate system, especially in cases that the groundwater level is shallow. The dynamic feedback from groundwater to land surface processes can introduce a second-order impact on the groundwater table and travel times. A fully coupled system, based on a mixed form of the Richards equation, is more realistic than the one-way system [1]. However, fully coupled

model constantly suffers from expensive computational burden, which limits its applicability in large-scale real-world models. It also introduces extra parameters, which are essentially unknown at the catchment scale. Due to the above reasons, we didn't apply the fully coupled modeling approach in this study.

We think the current one-way coupling is appropriate in terms of investigating groundwater resources, because: (1) the current modeling method, although less accurate than the fully-coupled model, is computationally more efficient. It reduces the computational demand of this study. Besides, the Lagrangian particle tracking in the 3D model is computationally very expensive (total computational time is 20 days using 40 cores on a super computer for the ensemble simulations). (2) This method facilitates the use of currently established land surface models in the HOKLIM project to investigate large-scale groundwater levels and travel times, without re-establishing all the models [2,3]. (3) This one-way coupling method has been widely used by many other researchers [4,5,6].

We also agree with the reviewer that the limitation of the current method should be discussed in the manuscript. Accordingly, we modified the discussion section to include the following information: *"A potential limitation of the current modeling framework lies in the simple one-way coupling approach that neglects the feedback from groundwater level change to the land surface processes. The change in groundwater table can alter the partitioning of water balances, which further exerts a second-order impact on the groundwater level and travel times. The fully coupled system, based on a mixed form of the Richards equation to solve unsaturated and saturated zone flow simultaneously, is more realistic than the one-way coupled system. However, fully coupled model consistently suffers from expensive computational burden, which limits its applicability in large scale real-world models. It also introduces extra parameters that are essentially unknown at the catchment scale. The current one-way coupling, although less accurate than the two-way coupling, is computationally more efficient considering the huge computational demand with the Lagrangian particle tracking. The current approach also facilitates the extension of currently established land surface models in the HOKLIM project for investigating large-scale groundwater levels and travel times with minimal additional effort. The one-way coupling approach has also been widely used by many other researchers [4,5,6]…"*

Another aspect that is not fully explained is the validation of the groundwater simulations. The authors touch very briefly this point by saying that mHM has been validated at the European scale in a previous paper, but what about this specific small catchment or the larger, but still small compared to the European scale, Thuringian

catchment? And what about the groundwater model? At page 8, line 17 I read: "The post-calibrated values of the hydraulic conductivity in each geological unit obtained from a previous study are assigned to the corresponding geological layers of the mesh (Jing et al., 2018a). Meanwhile, a uniform porosity of 0.2 is assigned to each geological layers (Table 2)". In a previous work (Jing et al., 2018a) the authors presented a comparison between observed and simulated heads at a number of observation wells (Figure 5 and related text on section 3.2.5) and for a number of recharge scenarios. The analysis is based on 400 calibrated Monte Carlo realizations and I am wondering if the authors used all the 400 realizations in the present work, or just one, in the latter case what was the criteria used to assign the hydraulic conductivity? Figure 5 of the previous paper shows apparently a good reproduction of the observed heads, but what puzzled me is that the standard deviation of the error is 4.6 m, a rather large portion of the variation presented in this manuscript as an effect of climate change.

Response: We thank the reviewer for these insightful comments. We agree with the reviewer that the validation of groundwater model is very important, and will add more information on the validation of groundwater model in the revised manuscript.

The mHM model has been validated both on the European scale and the catchment scale. The validation result of mHM model on the Thuringian basin is included in previous publications [7,8].

The groundwater model was validated using the observed groundwater levels in many monitoring wells. For the steady state model, a long-term average of observed groundwater levels is compared to the simulation results. The simulation results show a good correspondence to the observations (Fig. 1).

[Figure]

Fig. 1 Validation of groundwater model

The current hydraulic conductivity values for groundwater model are randomly sampled from the 400 parameter sets [9]. The reviewer pointed out that parameter values and its uncertainty can be critical to study results, with which we fully agree. In order to further investigate the parametric uncertainty in groundwater model, we expanded the current parameter set from only one set to 80 different sets. Those 80 parameter sets are all compatible with the groundwater level observations. The 80 parameter sets are randomly sampled from the parameter dataset used in our previous study [9].

The standard deviation of the residuals between observed and simulated groundwater table is 3 to 5 m, which seems to be large. However, the topography is complex in the study area, and the groundwater level difference between the highest and lowest monitoring wells is around 220 m. The CV (coefficient of variation) of the residuals is 2.09%, which is quite low, demonstrating a good reproduction of groundwater dynamics.

According to these important comments, we modified the manuscript to include simulations using many realizations of hydraulic conductivity fields (80 members in total). We also expanded the study to investigate the parametric uncertainty in groundwater model associated with 80 different hydraulic conductivity fields. We presented a comprehensive uncertainty study in the revised manuscript.

The third issue I would like to comment is uncertainty. My impression is that combining a large number of GCM-RCM pairs introduced a large variability of the meteorological forcing and therefore to water levels and travel time distributions, which should be validated in some way. According to the presented analysis it is very difficult, if not impossible, to sort out unrealistic scenarios, or weight less them in the ensemble. On the other hand, uncertainty in the hydrological models is neglected, in particular that related to the groundwater model.

Response: This is another important observation by the reviewer. Substantial uncertainty persists about the impacts of climate change on mean precipitation from general circulation models (GCMs) [10, 11]. A large spread in projections occurs in many regions and variables due to a combination of variations in the climate sensitivity that determines the magnitude of the average global response, and large variations in the spatial patterns of change – particularly for precipitation [10]. It is a

widely-used approach to set up a large ensemble of GCMs and greenhouse-gas emissions scenarios to generate recharge projections, which is also the case of this study [4, 12, 13]. Besides, it is clear that when exploring the potential impact of climate change scenarios, ensemble spread provides some important, if incomplete, information about the range of plausible future climate changes. This uncertainty information significantly improves the usefulness of climate projection and impact information by (a) allowing policy makers to consider a plausible range of eventualities and (b) informing the appropriate use of uncertain climate projections [10]. We use a large ensemble of GCM-RCP combinations to assess the climate uncertainty, and we also use the summary statistics (e.g., mean and SD) to show the highest possible scenarios in the future.

We agree with the reviewer that the uncertainty in groundwater model should be considered. We modified the manuscript to include simulations using many realizations of hydraulic conductivity fields (80 members in total). We also expanded the study to investigate the parametric uncertainty in groundwater model associated with 80 different hydraulic conductivity fields. We also presented a comprehensive uncertainty study in the revised manuscript.

Overall I feel that the manuscript, although interesting and pleasant to read, cannot be accepted for publication in the present form. As it is the manuscript reads like an application of previously published modeling efforts, with little analysis of the results. However, I see value in what the author presented and I think that with some extra effort they may accommodate the above drawbacks, by explaining more the underlying hypotheses and limitations of the current analysis and improving and enriching the discussion of simulation results. With these changes I think this manuscript will be a valuable contribution to the community.

Response: We are modifying the manuscript thoroughly according to the reviewer's comments. We will post a revised manuscript soon for HESS.

Minor comments Page 9 line 6: The recharge seems to increase almost linearly with the temperature, which is strange considering the many nonlinearities involved in the infiltration process;

Response: Yes, but it only indicates that the spread in projected recharges increase in a near-linear way. If we look at the ensemble average of projected recharges, it shows a nonlinear relationship between the recharge and warming level (8.0%, 8.9%, and 7.2% for the 1.5, 2, and 3 degree).

We already stated this in the current manuscript: "*The ensemble simulations do not show a systematic relationship between the predicted change and the warming level, but they indicate an increased variability in predicted changes with the enhanced warming level from 1.5 to 3 degree.*'' (Page 1, Line 10-11)

Page 9, line 9: "The projected....." This seems to suggest that the number of combinations of GCM and RCM can be reduced, or even that similar results can be obtained by using only the GCM. Please elaborate a bit more.

Response: We use different pairs of GCMs and RCPs to provide the degree of spread in future climate projections. It is clear that when exploring the potential impact of climate change scenarios, ensemble spread provides some important, if incomplete, information about the range of plausible future climate changes.

We nonetheless fully agree with the reviewer that this point should be elaborated in detail. Accordingly, we added the following information into the manuscript*:" Nevertheless, differences in recharge induced by different RCPs can still be witnessed in Figure 3, indicating the necessity of considering multiple GCM/RCP combinations for providing a plausible range of predictive uncertainty."*

Page 14, line 17 and following: this sentence is vague. Have these shallow local flow pathways actually been observed in the simulations and how realistic are they?

Response: These shallow local flow pathways exist in areas with complex topography and dense drainage network. In areas with complex topography and dense drainage network, rising groundwater level may activate shallow groundwater flow paths and intensify shallow local flow pathways. This effect is evidenced by the simulation results. We modify the revised manuscript accordingly to clarify this point.

Page 15 line, 25: this disclaimer, saying that uncertainty may be even larger, since some uncertainty sources have not been considered is somewhat alarming because it casts doubts on the interpretation of the results.

Response: We agree with the reviewer that the more uncertainty sources, e.g., uncertainty in groundwater model, should be considered. We modified the manuscript to include simulations using many realizations of hydraulic conductivity fields (80 members in total). We also expanded the study to investigate the parametric uncertainty in groundwater model associated with 80 different hydraulic conductivity fields. In doing so, a more comprehensive coverage of multiple uncertainty sources is achieved. We presented a comprehensive uncertainty study in the revised manuscript.

Page 16, line 20: How can be that first-order effects of climate change are small and second-order effects are not negligible? If for not negligible you mean that they are however smaller than first-order effects, what is the reliability of their estimate considering the large uncertainty affecting these simulations? Please elaborate more

Response: In the context of this paragraph, the term "first-order effect" means the direct effect of climate change on recharge, and the term "second-order effect" means the effect on groundwater quantity and travel times introduced by the change of recharge. We want to deliver the information that although the relative change in recharge rate seems to be indistinctive, its further effect on groundwater levels and travel times can be significant according to our simulation results.

This original expression, as pointed out by the reviewer, may be somehow misleading. We modified this sentence as the following one: *"To summarize, climate change can significantly alter the quantity and travel time behavior of regional groundwater system through the modification of recharge, especially for the long term. Ensemble simulations indicate remarkable predictive uncertainties in regional groundwater quantity and travel times, which are introduced by both climate projection and groundwater model."*

**References**

1. Yeh PJF, Eltahir EAB (2005) Representation of water table dynamics in a land surface scheme: 1. Model development. J Clim 18(12):1861–1880

2. Thober, S., Kumar, R., Wanders, N., Marx, A., Pan, M., & Rakovec, O. (2018). Multi-model ensemble projections of European river floods and high flows at 1.5, 2, and 3 degrees global warming.

3. Marx, A., Kumar, R., Thober, S., Rakovec, O., Wanders, N., Zink, M., … Samaniego, L. (2018). Climate change alters low flows in Europe under global warming of 1.5, 2, and 3°C. *Hydrology and Earth System Sciences*, *22*(2), 1017–1032. http://doi.org/10.5194/hess-22-1017-2018

4. Jackson, C. R., Meister, R., & Prudhomme, C. (2011). Modelling the effects of climate change and its uncertainty on UK Chalk groundwater resources from an ensemble of global climate model projections. *Journal of Hydrology*, *399*(1–2), 12–28. http://doi.org/10.1016/j.jhydrol.2010.12.028

5. Pulido-Velazquez, M., Peña-Haro, S., García-Prats, A., Mocholi-Almudever, A. F., Henriquez-Dole, L., Macian-Sorribes, H., & Lopez-Nicolas, A. (2015). Integrated assessment of the impact of climate and land use changes on groundwater quantity and quality in the Mancha Oriental system (Spain). *Hydrology and Earth System Sciences*, *19*(4), 1677–1693. http://doi.org/10.5194/hess-19-1677-2015

6. Sutanudjaja, E. H., Van Beek, L. P. H., De Jong, S. M., Van Geer, F. C., & Bierkens, M. F. P. (2011). Large-scale groundwater modeling using global datasets: A test case for the Rhine-Meuse basin. *Hydrology and Earth System Sciences*, *15*(9), 2913–2935. http://doi.org/10.5194/hess-15-2913-2011

7. Heße, F., Zink, M., Kumar, R., Samaniego, L. & Attinger, S. Spatially distributed characterization of soil-moisture dynamics using travel-time distributions. *Hydrol. Earth Syst. Sci.* **21,** 549–570 (2017).

8. Jing, M. *et al.* Improved regional-scale groundwater representation by the coupling of the mesoscale Hydrologic Model (mHM v5.7) to the groundwater model OpenGeoSys (OGS). *Geosci. Model Dev.* **11,** 1989–2007 (2018).

9. Jing, M., Heße, F., Kumar, R., Kolditz, O., Kalbacher, T., & Attinger, S. (2019). Influence of input and parameter uncertainty on the prediction of catchment-scale groundwater travel time distributions. *Hydrology and Earth System Sciences*, *23*(1), 171–190. http://doi.org/10.5194/hess-23-171-2019

10.      McSweeney, C. F. & Jones, R. G. How representative is the spread of climate projections from the 5 CMIP5 GCMs used in ISI-MIP? *Clim. Serv.* **1,** 24–29 (2016).

11.      Bates, B. C., Kundzewicz, Z. W., Wu, S. & Palutikof, J. P. Climate Change and Water Technical Paper of the Intergovernmental Panel on Climate Change VI (IPCC, 2008).

12.      Tillman, F. D., Gangopadhyay, S. & Pruitt, T. Changes in groundwater recharge under projected climate in the upper Colorado River basin. 6968–6974 (2016). doi:10.1002/2016GL069714.

13.      Goderniaux, P., Brouyère, S., Wildemeersch, S., Therrien, R. & Dassargues, A. Uncertainty of climate change impact on groundwater reserves - Application to a chalk aquifer. *J. Hydrol.* **528,** 108–121 (2015).

---

## Author Comment (AC2) · 21 May 2019

**Assessing the response of groundwater quantity and travel time distribution to 1.5, 2 and 3 degrees global warming in a mesoscale central German basin**

Miao Jing[1,2], Rohini Kumar[1], Falk Heße[1], Stephan Thober[1], Oldrich Rakovec[1,3], Luis Samaniego[1], and Sabine Attinger[1,4]

[1]Department of Computational Hydrosystems, UFZ – Helmholtz Centre for Environmental Research, Permoserstr. 15, 04318 Leipzig, Germany
[2]Institute of Geosciences, Friedrich Schiller University Jena, Burgweg 11, 07749 Jena, Germany
[3]Czech University of Life Sciences, Faculty of Environmental Sciences, Prague, 169 00, Czech Republic
[4]Institute of Earth and Environmental Sciences, University of Potsdam, Karl-Liebknecht-Str. 24–25, 14476 Potsdam, Germany

**Correspondence:** Miao Jing (miao.jing@ufz.de); Falk Heße (falk.hesse@ufz.de)

**Abstract.** Groundwater is the biggest single source of high-quality fresh water worldwide, which is also continuously threatened by the changing climate. This paper is designed to investigate the response of regional groundwater system to the climate change under three global warming levels (1.5, 2, and 3 °C) in a central German basin (Nägelstedt). This investigation is conducted by deploying an integrated modeling workflow that consists of a mesoscale Hydrologic Model (mHM) and a fully-distributed groundwater model OpenGeoSys (OGS). mHM is forced by five general circulation models under three representative concentration pathways. The diffuse recharges estimated by mHM are used as outer forcings of the OGS groundwater model to compute changes in groundwater levels and travel time distributions. Simulation results  indicates that under future climate scenarios, groundwater recharges and levels are expected to increase slightly. Meanwhile, the mean travel time is expected to decrease compared to the historical average. However, the ensemble simulations do not all agree on the sign of relative change. The ensemble simulations do not show a systematic relationship between the projected change and the warming level, but they indicate an increased variability in projected changes with the enhanced warming level from 1.5 to 3 °C.  The predictive uncertainties related to climate projections are more pronounced than that related to hydraulic conductivity fields. Our results imply that 
[revised manuscript text omitted]

---

## Referee Comment (RC2) · Anonymous Referee #2 · 27 Aug 2019

The authors of the manuscript propose a combined three-level modeling approach to investigate the influence of climate change on the groundwater levels and groundwater travel time in a small agricultural watershed in central Germany. They use 5 different global circulation models, which provide climate data for mesoscale Hydrologic Model mHM. In turn, mHM predicts values of groundwater recharge, which are in turn used as input in a 3D saturated groundwater flow model implemented in OpenGeoSys. Thus, their work is a valuable contribution to the development of comprehensive modeling approaches describing hydrologial systems. This type of analysis is much needed in

view of the discussion on the possible effects of global warming. The main finding is that the influence of climate change on the groundwater travel time is more pronounced than the influence on groundwater levels.

I agree with the comments of the first reviewer, who pointed out important limitations of the manuscript. They are related to (i) neglecting of unsaturated zone processes and the influence of shallow groundwater table on surface hydrology, (ii) use of coarse-grid model for calculating recharge rates, (iii) other possible sources of uncertainty, besides the differences between climate models. In the revised version, these issues were addressed by providing additional simualtions and extended discussion.

My general comments related to the current version of the manuscript are as follows: 1. I would like to see more information about the actual values of recharge and recharge/precipitation ratio in different scenarios. Does the recharge change proportionally to the precipitation in all scenarios, or maybe there were some nonlinear effects, such as those mentioned by the authors on page 3, lines 2-4?

2. What was the spatial variability of recharge obtained from mHM ? Even using 5x5 km grid you should see some differences in the watershed area. Was the degree of variability similar in all scenarios?

3. On page 17, lines 10-15 the authors mention that their model is able to simulate correctly the appearance of additional groundwater discharge zones when the water table rises, as shown in Fig.9. This should be explained in more detail. How is this kind of boundary condition treated in OpenGeoSys? Is it possible that groundwater heads in the top layer of cells are above the ground level ? It would be nice to see actual model results supporting the concept shown in Fig. 9.

Technical correction: Page 5, last line "C" after "degree" symbol seems redundant.

---

## Author Response (AR1)

**Responses to Referee Review 1**

We thank the referee reviewer for his comprehensive and insightful comments. Our responses to the reviewers' comments are given below. The original comments from referee reviewer 1 were marked with blue color, and our response in black. The page and line numbers in our responses refer to those in the **marked copy** of the revised texts.

**General comments**

From the description provided in the methodology and materials section I argue that mHM and OGS, the mesoscale hydrological model and the groundwater model are decoupled. In fact, at line 13 of page 5 I read "The projected recharge from mHM calculations are fed to the groundwater model OpenGeoSys (OGS) for the assessment of groundwater quantity and travel time distribution." From this short description I concluded that vertical fluxes, including those of the unsaturated zone are modeled with mHM and that the resulting "deep" infiltration is used as recharge (i.e., as boundary condition) for OGS. In doing that the processes of infiltration and flow inside the aquifer are decoupled. This may be reasonable when the water table is deep, but I am wondering what is the impact of this assumption on the simulations showing significant rises of the water table. In this situation decoupling leads to significant errors as the water table gets close to the ground surface. This is relevant both for water levels and the following travel time analysis. The authors provide only limited information on this important point and do not discuss its implications in term of representativity of the model. Decoupling these two processes is a great advantage from a modeling point of view, but I am not sure it can be actually introduced at least in the scenarios showing the largest increase of groundwater levels.

Response:

We fully agree with the reviewer that groundwater dynamics can alter land surface response and feedback to the climate system, especially in cases that the groundwater level is shallow. The dynamic feedback from groundwater to land surface processes can introduce a second-order impact on the groundwater table and travel times. A fully coupled system, based on a mixed form of the Richards equation, is more realistic than the one-way system [1]. However, fully coupled model constantly suffers, among other things, from expensive

computational burden, which limits its applicability in large-scale real-world models. It also introduces data burden involving several extra parameters, which are essentially unknown at the catchment scale. Due to the above reasons, we didn't apply fully coupled system in this study.

Following previous studies [4,5,6], we adopted here the one-way coupling between the (near) surface hydrologic and groundwater models for investigating groundwater resources. We have successfully demonstrated the utility of this model for adequately capturing the observed behavior of groundwater levels across the study basin [8]. The current modeling method, although less accurate than the fully-coupled model, is computationally more efficient – allowing us to understand the first order control of climatic variability on groundwater characteristics (groundwater level and travel times). Notably  the applied Lagrangian particle tracking in the 3D groundwater model is computationally very expensive. The computational time performing a single model run is around 20 days using 40 cores on a super computer facility.  Our analyses make use of large hydro-climatic ensemble scenarios (based on the multiple GCMs/RCPs and a hydrologic model, mHM combinations) – thereby accounting for the input uncertainties to the driving groundwater model, which is particularly important and recommendable for any climate change impact assessment studies [4,12,13,14]. Accordingly the one-way coupling method used here is a practical choice, allowing us to perform the large ensemble scenarios with reasonable computational resources (and time).

Nevertheless, we also agree with the reviewer that the limitation of the current method should be discussed in the manuscript. Accordingly, we modified the discussion section to include the following information (please check **l.26-35, p.18**, and **l.1-4, p.19**): *"A potential limitation of the current modeling framework lies in the simple one-way coupling approach that neglects the feedback from groundwater level change to the land surface processes. The change in groundwater table can alter the partitioning of water balances, which further exerts a second-order impact on the groundwater level and travel times. The fully coupled system, based on a mixed form of the Richards equation to solve unsaturated and saturated zone flow simultaneously, is more realistic than the one-way coupled system. However, fully coupled model consistently suffers from expensive computational burden, which limits its applicability in large scale real-world models. It also introduces extra parameters that are essentially unknown at the catchment scale. The current one-way coupling, although less accurate than the two-way coupling, is computationally more efficient – allowing us to understand the first order control of climatic variability on groundwater*

*characteristics (groundwater level and travel times). Notably, the applied Lagrangian particle tracking in the 3D groundwater model is computationally very expensive. Moreover, we have successfully demonstrated the utility of this model for adequately capturing the observed behavior of groundwater levels across the study basin. Accordingly, the one-way coupling method used here is a practical choice, allowing us to perform the large ensemble scenarios with reasonable computational resources (and time)."*

Another aspect that is not fully explained is the validation of the groundwater simulations. The authors touch very briefly this point by saying that mHM has been validated at the European scale in a previous paper, but what about this specific small catchment or the larger, but still small compared to the European scale, Thuringian catchment? And what about the groundwater model? At page 8, line 17 I read: "The post-calibrated values of the hydraulic conductivity in each geological unit obtained from a previous study are assigned to the corresponding geological layers of the mesh (Jing et al., 2018a). Meanwhile, a uniform porosity of 0.2 is assigned to each geological layers (Table 2)". In a previous work (Jing et al., 2018a) the authors presented a comparison between observed and simulated heads at a number of observation wells (Figure 5 and related text on section 3.2.5) and for a number of recharge scenarios. The analysis is based on 400 calibrated Monte Carlo realizations and I am wondering if the authors used all the 400 realizations in the present work, or just one, in the latter case what was the criteria used to assign the hydraulic conductivity? Figure 5 of the previous paper shows apparently a good reproduction of the observed heads, but what puzzled me is that the standard deviation of the error is 4.6 m, a rather large portion of the variation presented in this manuscript as an effect of climate change.

Response: We thank the reviewer for these insightful comments. We agree with the reviewer that the validation of groundwater model is very important, and will add more information on the validation of groundwater model in the revised manuscript.

The mHM model has been validated both on the European scale and the catchment scale. The validation result of mHM model on the Thuringian basin is included in previous publications [7,8].

The groundwater model was validated using the observed groundwater levels in many monitoring wells. For the steady state model, a long-term average of observed groundwater levels is compared to the

simulation results. The simulation results show a good correspondence to the observations (Fig. 1).

[Figure]

Fig. 1 Validation of groundwater model

The current hydraulic conductivity values for groundwater model are randomly sampled from the 400 parameter sets [9]. The reviewer pointed out that parameter values and its uncertainty can be critical to study results, with which we fully agree. In order to further investigate the parametric uncertainty in groundwater model, we expanded the current parameter set from only one set to 80 different sets. Those 80 parameter sets are all compatible with the groundwater level observations. The 80 parameter sets are randomly sampled from the parameter dataset used in our previous study [9].

The standard deviation of the residuals between observed and simulated groundwater table is 3 to 5 m, which seems to be large. However, the topography is complex in the study area, and the groundwater level difference between the highest and lowest monitoring wells is around 220 m. The CV (coefficient of variation) of the residuals is 2.09%, which is quite low, showing a good reproduction of groundwater dynamics.

According to these important comments, we modified the manuscript to include simulations using many realizations of hydraulic conductivity fields (80 members in total). We also expanded the study to investigate the parametric uncertainty in groundwater model associated with 80 different hydraulic conductivity fields. We presented a comprehensive uncertainty study in the revised manuscript.

The third issue I would like to comment is uncertainty. My impression is that combining a large number of GCM-RCM pairs introduced a large variability of the meteorological forcing and therefore to water levels and travel time distributions, which should be validated in some way. According to the presented analysis it is very difficult, if not impossible, to sort out unrealistic scenarios, or weight less them in the ensemble. On the other hand, uncertainty in the hydrological models is neglected, in particular that related to the groundwater model.

Response: This is another important observation by the reviewer. Substantial uncertainty persists regarding the impacts of climate change on mean precipitation based on the general circulation models (GCMs) under historical and future pathways scenarios [10, 11]. A large spread in projections occurs in many regions and variables due to a combination of variations in the climate sensitivity that determines the magnitude of the average global response, and large variations in the spatial patterns of change – particularly for precipitation [10]. It is a widely used (and recommended) approach to use a large ensemble of GCMs and greenhouse-gas emissions scenarios to account for the inherent uncertainties in impact assessment studies [4, 12, 13]. Besides, it is clear that when exploring the potential impact of climate change scenarios, ensemble spread provides some important, if incomplete, information about the range of plausible future climate changes. This uncertainty information significantly improves the usefulness of climate projection and impact information by (a) allowing policy makers to consider a plausible range of eventualities and (b) informing the appropriate use of uncertain climate projections [10]. We note that the underlying five GCM datasets selected here is a subset of a large CMIP5 archive; covering a range of 0.55 of the uncertainty of the entire CMIP5 ensemble for precipitation and 0.75 for temperature (McSweeney and Jones, 2016). Also these models are bias-corrected with observational datasets in the historical period using the trend-preserving bias correction (see Hempel et al., 2013 for more details); and have been the basis for a large number of impact assessment studies   (ISI-MIP; https://www.isimip.org). Accordingly we did not perform any sub-selection and use all available ensemble of GCM-RCP combinations to assess the climate uncertainty, and provided the summary statistics (e.g., mean and SD) to show the possible future scenarios for groundwater resources across the Study area.

In the recent studies, we have evaluated the suitability of GCMs based hydrologic simulations using available streamflow datasets [Samaniego et al., 2018; Thober et al., 2018; Marx et al., 2018]. We agree with the reviewer that the uncertainty in groundwater model should be considered. Indeed in our previous study [9], we performed a detailed investigation on the issue of uncertainty. We modified the manuscript to include simulations using many realizations of hydraulic conductivity fields (80 members in total) – key source of uncertainty in groundwater models. We expanded the study to investigate the parametric uncertainty in groundwater model associated with 80 different hydraulic conductivity fields and contrasted the results with arising from the climate changes. We also presented a comprehensive uncertainty study in the revised manuscript (l.25-30. p.7, l.5-10, p.16, and l.1-10, p.17).

Overall I feel that the manuscript, although interesting and pleasant to read, cannot be accepted for publication in the present form. As it is the manuscript reads like an application of previously published modeling efforts, with little analysis of the results. However, I see value in what the author presented and I think that with some extra effort they may accommodate the above drawbacks, by explaining more the underlying hypotheses and limitations of the current analysis and improving and enriching the discussion of simulation results. With these changes I think this manuscript will be a valuable contribution to the community.

Response: With the responses provided above and the proposed modifications to the revised manuscript, we hope that reviewer may appreciate our work, and will overall find it as a valuable contribution.

Minor comments Page 9 line 6: The recharge seems to increase almost linearly with the temperature, which is strange considering the many nonlinearities involved in the infiltration process;

Response: Yes, but it only indicates that the spread in projected recharges increase in a near-linear way. If we look at the ensemble average of projected recharges, it shows a nonlinear relationship between the recharge and warming level (8.0%, 8.9%, and 7.2% for the 1.5, 2, and 3 degree).

Page 9, line 9: "The projected....." This seems to suggest that the number of combinations of GCM and RCM can be reduced, or even that

similar results can be obtained by using only the GCM. Please elaborate a bit more.

Response: See our response above.

We however fully agree with the reviewer that this point should be elaborated in detail. Accordingly, we added the following information into the manuscript*:" Nevertheless, differences in recharge induced by different RCPs can still be witnessed in Figure 3, indicating the necessity of considering multiple GCM/RCP combinations for providing a plausible range of predictive uncertainty."* (l.11-12, p.11)

Page 14, line 17 and following: this sentence is vague. Have these shallow local flow pathways actually been observed in the simulations and how realistic are they?

Response: These shallow local flow pathways exist in areas with complex topography and dense drainage network. In those regions, rising groundwater level may activate shallow groundwater flow paths and intensify shallow local flow pathways. This effect is evidenced by the simulation results. We modify the revised manuscript accordingly to clarify this point.

Page 15 line, 25: this disclaimer, saying that uncertainty may be even larger, since some uncertainty sources have not been considered is somewhat alarming because it casts doubts on the interpretation of the results.

Response: We agree with the reviewer that the more uncertainty sources, e.g., uncertainty in groundwater model, should be considered. We modified the manuscript to include simulations using many realizations of hydraulic conductivity fields (80 members in total). We also expanded the study to investigate the parametric uncertainty in groundwater model associated with 80 different hydraulic conductivity fields. We now present a comprehensive uncertainty study in the revised manuscript (l.25-30. p.7, l.5-10, p.16, and l.1-10, p.17).

Page 16, line 20: How can be that first-order effects of climate change are small and second-order effects are not negligible? If for not negligible you mean that they are however smaller than first-order effects, what is the reliability of their estimate considering the large uncertainty affecting these simulations? Please elaborate more

Response: In the context of this paragraph, the term "first-order effect" means the direct effect of climate change on recharge, and the term "second-order effect" means the effect on groundwater quantity and travel times introduced by the change of recharge. We want to deliver the information that although the relative change in recharge rate seems to be indistinctive, its further effect on groundwater levels and travel times can be significant according to our simulation results.

This original expression, as pointed out by the reviewer, may be somehow misleading. We modified this sentence as the following one: *"To summarize, climate change can significantly alter the quantity and travel time behavior of regional groundwater system through the modification of recharge, especially for the long term. Ensemble simulations indicate a remarkable predictive uncertainty in regional groundwater quantity and travel times, which is introduced by both climate projection and groundwater model."* (l.3-8, p.20)

**Responses to Referee Review 2**

We are grateful to the second referee reviewer for his/her comprehensive and insightful comments. Our responses to the reviewers' comments are given below. The original comments from the referee reviewer were marked with blue color, and our response in black. The page and line numbers in our responses refer to those in the **marked copy** of the revised texts.

General comments

The authors of the manuscript propose a combined three-level modeling approach to investigate the influence of climate change on the groundwater levels and groundwater travel time in a small agricultural watershed in central Germany. They use 5 different global circulation models, which provide climate data for mesoscale hydrologic Model mHM. In turn, mHM predicts values of groundwater recharge, which are in turn used as input in a 3D saturated groundwater flow model implemented in OpenGeoSys. Thus, their work is a valuable contribution to the development of comprehensive modeling approaches describing hydrological systems. This type of analysis is much needed in view of the discussion on the possible effects of global warming. The main finding is that the influence of climate change on the groundwater travel time is more pronounced than the influence on groundwater levels.

We agree with the comments of the first reviewer, who pointed out important limitations of the manuscript. They are related to (i) neglecting of unsaturated zone processes and the influence of shallow groundwater table on surface hydrology, (ii) use of coarse-grid model for calculating recharge rates, (iii) other possible sources of uncertainty, besides the differences between climate models. In the revised version, these issues were addressed by providing additional simulations and extended discussion.

Response:

Thank you very much for your overall evaluation of our study. We will revise the manuscript carefully based on your comments.

My general comments related to the current version of the manuscript are as follows: 1. WE would like to see more information about the actual values of recharge and recharge/precipitation ratio in different scenarios. Does the recharge change proportionally to the precipitation in all scenarios, or maybe there were some nonlinear effects, such as those mentioned by the authors on page 3, lines 2-4?

Response: Thank you so much for these important observations. We fully agree with the reviewer that the actual value and ratio (as a proportion of precipitation) of recharge are critical to the understanding of climate effect. These behaviors are shown in Figure 1. We can see that the actual annual recharge rates are between 100 mm to 145 mm depending on different climate scenarios and warming levels. We can also observe that the change in recharge rate is not proportional to that in precipitation. For example, the recharge ratio in GFDL-ESM2M increases from 0.178 to 0.212 following the increase of warming levels. Conversely, the recharge ratio in HadGEM2-ES decreases slightly in 3 degree warming. This phenomenon indicates a non-linear relationship between the changes in recharge and precipitation depending on different climate models.

[Figure]

**Figure 1 Actual recharge rates and recharge ratio (as a proportion of precipitation) under different warming levels.**

2. What was the spatial variability of recharge obtained from mHM? Even using 5x5 km grid you should see some differences in the watershed area. Was the degree of variability similar in all scenarios?

Response: This is an important observation. To answer this question, we take a close look at the spatial pattern of recharges in different climate scenarios. Specifically, we find that the spatial patterns of projected recharges appear to be very similar among each other (Figure 2). However, the relative changes in recharges are spatially heterogeneous (Figure 3). This spatial variability can be attributed to the heterogeneous topography and land use. Alternatively speaking, the degree of changes depends on the local topographic, morphologic, and hydraulic properties of soils. This shows the importance of deploying a spatially distributed hydrological model in projecting regional hydrological responses. We modified the manuscript accordingly (please check l.1-3, p.16).

[Figure]

**Figure 2  Spatial distributions of projected recharges under 1.5 degree warming.**

[Figure]

**Figure 3 Spatial distributions of relative changes in projected recharges under 1.5 degree warming.**

3. On page 17, lines 10-15 the authors mention that their model is able to simulate correctly the appearance of additional groundwater discharge zones when the water table rises, as shown in Fig.9. This should be explained in more detail. How is this kind of boundary condition treated in OpenGeoSys? Is it possible that groundwater heads in the top layer of cells are above the ground level? It would be nice to see actual model results supporting the concept shown in Fig. 9.

Response: Thank you so much for your insights. We would like to clarify that Figure 9 in the manuscript is a conceptual graph that shows a possible consequence of the increased groundwater levels (Havril et al., 2018; Kaandorp et al., 2018; Toth, 1963). The current groundwater model is based on predefined geometry of stream network, and is not able to simulate the appearance of additional groundwater discharge zones. We discuss the possible consequences of a rising groundwater level, especially in areas where the groundwater depth is shallow.

Many past studies have demonstrated that the rise of groundwater level in shallow groundwater aquifers will lead to the activation of additional discharge paths (Havril et al., 2018; Kaandorp et al., 2018; Toth, 1963). In the current model, the discharge zones (streams) are predefined and do not change in the simulations. Specifically, a fixed head boundary is assigned to the main perennial streams including one mainstream and three tributaries. From our simulations, we find that the large changes in groundwater levels happen at hilly areas, whereas changes in central lowlands are not as significant as those in hilly areas. We carefully checked all simulation results to ensure that the groundwater levels are all below the ground levels.

Note that this study is not designed to investigate the change in discharge zones under the climate change. Rather, it is designed to investigate the trend and the predictive uncertainty in the quantity and travel times of a regional groundwater system using ensemble simulations. To avoid possible misunderstanding and misinterpretation of Figure 9, we deleted it in the revised revision. We also modified the relevant discussions to avoid potential misinterpretations.

4. Technical correction: Page 5, last line "C" after "degree" symbol seems redundant.

Response: Modified as proposed.

[revised manuscript text omitted]

---

## Referee Report (RR1)

**Review of "Assessing the response of groundwater quantity and travel time distribution to 1.5, 2 and 3 degrees global warming in a mesoscale central German basin" by Jung et al. (2020)**

The objective of this manuscript is to assess the impacts of climate change on groundwater level and travel time distribution using numerical modelling. In the modelling framework, a distributed groundwater flow model OpenGeoSys (OGS) is driven by recharge simulated by mesoscale Hydrologic Model (mHM). Using this model, the authors estimate the response of groundwater level and travel time to different level of warming due to climate change in a Central German basin.

This is the second round of review of this manuscript. I went through the reviewer comments from the previous round. I believe that the authors have done a good job addressing the relevant issues.

In general, the manuscript is well-written, organised, and reads well. I feel that the manuscript could be accepted for publication after a few updates that I would like to suggest:

1) Table 2: Please provide the full name of the geologic units e.g., Middle Keuper (km).

2) Figure 3: Please provide the legend for the coloured markers in the plot.

3) Section 5.1 and 5.2: Section 5.1 provides the impacts of warming on recharge. Subsequently, Section 5.2 provides the resulting change in groundwater level. In Section 5.1, please provide a short discussion on the reasons of the changes in recharge simulated by mHM for different RCPs. Is it increased precipitation estimated by the GCMs in the study are over the simulation period? Similarly, in Section 5.2, please provide a clearer connection between the changes in groundwater level and simulated recharge for different RCPs.

4) Figure 5a: Groundwater level (m). m above what (reference level)? MSL? Please mention that in the figure.

---

## Author Response (AR2)

**Response Letter to Referee Report 3**

We thank the referee reviewer for his comprehensive and insightful comments. Our responses to the reviewers' comments are given below. The original comments from referee reviewer 1 were marked with black color, and our response in blue. The page and line numbers in our responses refer to those in the revised texts.

The objective of this manuscript is to assess the impacts of climate change on groundwater level and travel time distribution using numerical modelling. In the modelling framework, a distributed groundwater flow model OpenGeoSys (OGS) is driven by recharge simulated by mesoscale Hydrologic Model (mHM). Using this model, the authors estimate the response of groundwater level and travel time to different level of warming due to climate change in a Central German basin.

This is the second round of review of this manuscript. I went through the reviewer comments from the previous round. I believe that the authors have done a good job addressing the relevant issues.

In general, the manuscript is well-written, organised, and reads well. I feel that the manuscript could be accepted for publication after a few updates that I would like to suggest:

1) Table 2: Please provide the full name of the geologic units e.g., Middle Keuper (km).

Response: Thank you for your comments. We changed accordingly (see the updated Table 2).

2) Figure 3: Please provide the legend for the coloured markers in the plot.

Response: Changed accordingly (see the updated Figure 3).

3) Section 5.1 and 5.2: Section 5.1 provides the impacts of warming on recharge. Subsequently, Section 5.2 provides the resulting change in groundwater level. In Section 5.1, please provide a short discussion on the reasons of the changes in recharge simulated by mHM for different RCPs. Is it increased precipitation estimated by the GCMs in the study are over the simulation period? Similarly, in Section 5.2, please provide a clearer connection between the changes in groundwater level and simulated recharge for different RCPs.

Response: Thank you for your insightful comments. The reasons for the changes in recharge simulated by mHM for different RCPs are as follows:

''Under the same rising temperature and GCM, projected recharges still vary for different RCPs. The scale and distribution of precipitation change respond not only to temperature rise but also to employed RCPs (Mitchell et al., 2016; Thober et al., 2018). This is introduced by non-$CO_2$ forcing and the dependence of precipitation sensitivity to emission scenarios (Mitchell et al., 2016).'' (Page 10, Line 12-15)

The increased precipitation estimated by GCMs is not over the simulation period. Rather, it varies over the simulation period. In this study, we take the long-term mean to show the long-term tendency (Page 6, Line 6-7).

We also updated Figure 6 with the changes in groundwater levels with different RCPs. We also illustrate the relation between changes in groundwater levels and recharges for different RCPs:

''Estimated groundwater levels also respond to different RCPs, which is attributed to the RCP-dependent precipitation and its modification on recharge. Alternatively speaking, the uncertainty in RCPs is ultimately propagated from the RCP-dependent precipitation to the groundwater levels." (Page 12, Line 12-15)

4) Figure 5a: Groundwater level (m). m above what (reference level)? MSL? Please mention that in the figure.

Response: Agreed. We changed the caption of Figure 5 as follows:

[revised manuscript text omitted]